# Differentially Private Linear Regression via Medians

## Abstract

Linear regression is one of the simplest machine learning tasks. Despite much work, differentially private linear regression still lacks effective algorithms. We propose a new approach based on a multivariate extension of the Theil-Sen estimator. The theoretical advantage of our approach is that we do not directly rely on noise addition, which requires bounding the sensitivity. Instead we compute differentially private medians as a subroutine, which are more robust. We also show experimentally that our approach compares favourably to prior work.

## 1 Introduction

**Background & Motivation** Differential Privacy [DMNS06] is a standard for ensuring that the output (i.e., trained model) of a machine learning system does not leak sensitive details about its input (i.e., training data, which could contain private information about individual people). Differentially private machine learning has been the topic of considerable research, both theoretical and empirical, and is also used in practice [MT22].

Arguably, the simplest machine learning task is linear regression. That is, we are given a dataset $(x_1, y_1), (x_2, y_2), \cdots, (x_n, y_n) \in \mathbb{R}^d \times \mathbb{R}$ and our goal is to fit a linear model of the form $y_i \approx \langle \theta, x_i \rangle$ for some $\theta \in \mathbb{R}^d$. More precisely, ordinary least squares linear regression minimizes the squared error $\sum_i^n (\langle \theta, x_i \rangle - y_i)^2$. This objective corresponds to assuming that the errors (i.e., the deviations from a perfect linear relationship) are Gaussian. This objective is particularly nice, as it has a closed-form solution: $\theta = (X^T X)^{-1} X^T y$, where $X = (x_1, x_2, \ldots, x_n)^T \in \mathbb{R}^{n \times d}$ and $y = (y_1, y_2, \ldots, y_n)^T \in \mathbb{R}^n$.

Given the practical importance of linear regression, there has been a lot of work on differentially private linear regression. (We discuss the related work in more detail in Section 1.3.) However, these prior works all suffer from the same limitation: To guarantee differential privacy they add noise to some quantity – either to the raw data $X$ and $y$, to the sufficient statistics $X^T X$ and $X^T y$, or to the gradients $\sum_i x_i \cdot (\langle \theta, x_i \rangle - y_i)$ encountered when optimizing the least squares objective. This noise addition approach requires bounding the sensitivity, which essentially means we must provide a priori bounds on $\|x_i\|$ and $|y_i|$ or, rather, we must scale/clip the quantities of interest to enforce these bounds. The clipping hyperparameter induces a harsh privacy-utility tradeoff: If the bounds are loose, we must add more noise than necessary. If the bounds are too tight, the clipping distorts the data. This raises the question: *Can we perform differentially private linear regression in a way that is (nearly) agnostic to the sensitivity?*

**Inspiration for Our Approach** To gain some intuition, consider the even simpler task of mean estimation, i.e., computing the average $\frac{1}{n} \sum_i^n x_i$. Here we face the same difficulty in terms of clipping the data to bound the sensitivity. Numerous approaches to mean estimation have been studied [e.g.: KV17; BS19; KSU20; BDKU20; LKKO21; LKO21].

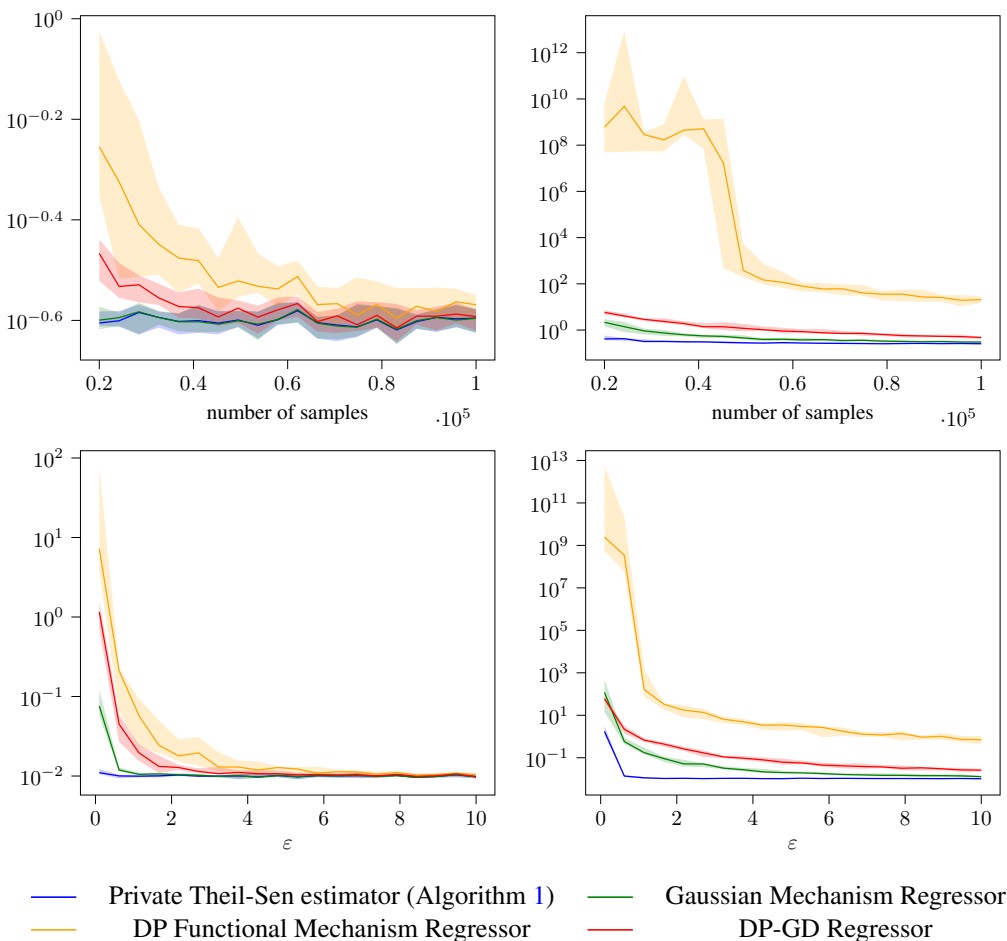

Figure 1: Comparison of DP linear regression algorithms. Mean square error (i.e., $\mathbb{E}[(\langle\hat{\theta}, x\rangle - y)^2]$ on vertical axis in logarithmic scale) as a function of the number of samples (i.e., $n$ on horizontal axis) for dimensions $d = 10$ (left) and $d = 30$ (right); and a function of $\varepsilon$ for $n = 10^5$ and dimensions $d = 10$ (left) and $d = 30$ (right). The line show the median and the semitransparent shadow shows the 0.1 and 0.9 quantiles of the error; values are computed over 20 runs. Privacy parameters are $\varepsilon = 1$ and $\delta = 10^{-6}$; and $\ell = 1$. Data is synthetic, see Section 2.1 for details.

One way to sidestep this sensitivity issue is to look at the median instead of the mean. Under reasonable distributional assumptions, the median is a good approximation to the mean, with the advantage that the sensitivity of the median is usually much lower than the mean. Thus the median can be a good tool for differentially private mean estimation.

The key innovation of our approach is to carry this median-instead-of-mean idea over to the setting of linear regression. But this is far from straightforward – we are interested in the multidimensional setting and even defining a multi-dimensional median is nontrivial.

We draw further inspiration from the literature on robust statistics – intuitively, the median is a robust replacement for the mean. In particular, the Theil-Sen estimator [The50; Sen68] uses the median to perform robust *simple* linear regression (i.e., $d = 1$). Indeed, a differentially private Theil-Sen estimator has been studied by Dwork and Lei [DL09] and Alabi, McMillan, Sarathy, Smith, and Vadhan [AMSSV22]. We extend this to multivariate linear regression using a variant of the (non-private) approach of Dang, Peng, Wang, and Zhang [DPWZ08].

---
**Algorithm 1** Private efficient multivariate Theil-Sen estimator.
---
1: **Input:** $(x_1, y_1), (x_2, y_2), \ldots, (x_n, y_n) \in \mathbb{R}^d \times \mathbb{R}$.
2: **Parameters:** Privacy parameter $\varepsilon > 0$. Number of partitions $\ell \geq 1$. Output range $\mathcal{R} \subset \mathbb{R}$.
3: Let $m = \lfloor n/d \rfloor$.
4: Initialize an empty multiset $\Theta \subset \mathcal{R}^d$.
5: **for** $k \in [\ell]$ **do**      ▷ Generate $\ell \cdot m$ subproblems $S_{j,k}$ such that each input appears in at most $\ell$.
6:      Randomly choose $m$ disjoint sets $S_{1,k}, S_{2,k}, \ldots, S_{m,k} \subset [n]$ each of size $d$.
7:      **for** $j \in [m]$ **do**
8:          Compute $\theta_{j,k} \in \mathbb{R}^d$ such that $\langle \theta_{j,k}, x_i \rangle = y_i$ for all $i \in S_{j,k}$.
9:          Project $\theta_{j,k} \in \mathbb{R}^d$ into $\tilde{\theta}_{j,k} \in \mathcal{R}^d$ – i.e., $\tilde{\theta}_{j,k} = \arg\min_{\tilde{\theta} \in \mathcal{R}^d} \|\tilde{\theta} - \theta_{j,k}\|$.
10:          Add $\tilde{\theta}_{j,k}$ to $\Theta$.
11:      **end for**
12: **end for**
13:                          ▷ Compute an approximate median $\hat{\theta} \in \mathcal{R}^d$ of the set $\Theta$ in a DP manner.
14: **for** $i \in [d]$ **do**   ▷ Independently sample $i$-th coordinate of $\hat{\theta}$ using the exponential mechanism.
15:      Sample $\hat{\theta}_i \in \mathcal{R}$ with probability proportional to

$$\mathbb{P}[\hat{\theta}_i] \propto \exp\left(-\frac{\varepsilon}{2\ell d} \max\left\{\left|\left\{\theta \in \Theta : \theta_i < \hat{\theta}_i\right\}\right|, \left|\left\{\theta \in \Theta : \theta_i > \hat{\theta}_i\right\}\right|\right\}\right).$$

16: **end for**
17: **return** $\hat{\theta}$.
---

## 1.1  Our Algorithm

Our private linear regression algorithm is described in Algorithm 1. We proceed with some remarks about our algorithm.

The high-level idea of the Theil-Sen estimator is that, rather than trying to solve the global objective (i.e., $\min_\theta \sum_i^n (\langle \theta, x_i \rangle - y_i)^2$), we solve $\ell \cdot m$ subproblems and then combine these solutions into a single solution via a median. Each subproblem consists of a subset of $d$ out of $n$ of the input points (which is enough to uniquely specify the weights $\theta_{j,k} \in \mathbb{R}^d$, assuming the $x_i$s are linearly independent).

The standard Theil-Sen estimator considers all $\binom{n}{d}$ possible subproblems. This is computationally prohibitive for realistic values of $n$ and $d$; hence we randomly select a subset of $\ell \cdot m$ subproblems. We will consider small numbers of repetitions, such as $\ell = 1$.

From a differential privacy perspective, changing one input point $(x_i, y_i)$ can change $\ell$ subproblems and hence $\ell$ elements of $\Theta$. If our method for computing the median is $(\varepsilon/\ell)$-differentially private with respect to changing one element of $\Theta$, then by group privacy it is $\varepsilon$-differentially private with respect to changing one input point $(x_i, y_i)$, as required.[1] This is a straightforward extension of the sample-and-aggregate framework [NRS07].

There are many ways to defube and compute a multivariate median (even non-privately). For simplicity, we compute a marginal median: we simply compute the univariate median for each coordinate – i.e., $\hat{\theta}_i \approx \text{median}_{\theta \in \Theta}\ \theta_i$ for each $i \in [d]$. Privately approximating the univariate median is a well-studied problem [NRS07; Smi08; DL09; Smi11; BNSV15; KV17; FS18; KLSU19; BS19; AD20; KLMNS20; GJK21; ABEC22]. We compute the median by a simple application of the exponential mechanism [MT07a]; although this doesn't achieve optimal asymptotic bounds, it performs remarkably well in practice. To be specific, following Smith [Smi11] and Feldman and Steinke [FS18], we sample each coordinate $\hat{\theta}_i$ from a probability distribution that decays exponentially with how far away it is from the median. This ensures that the overall algorithm satisfies $\varepsilon$-DP and is

---

[1]For simplicity, in this discussion, we restrict ourselves to pure differential privacy, but, to obtain better composition bounds in the high-dimensional setting, we will work with concentrated differential privacy [DR16; BS16] or approximate differential privacy [DKMMN06].

accurate under reasonable conditions. Each coordinate $\hat{\theta}_i$ is computed in a way that is $\varepsilon/d$-DP. Composing over the $d$ coordinates yields the final $\varepsilon$-DP bound.

Note that we restrict the range of the coordinates to $\mathcal{R} \subset \mathbb{R}$. This can either be an interval (e.g., $\mathcal{R} = [a, b]$) or a discrete set (e.g., $\mathcal{R} = \{a + (b - a) \cdot (i - 1)/r \; : \; i \in [r + 1]\}$). For the exponential mechanism to be well-defined, it is necessary to ensure that $\mathcal{R}$ has finite measure (i.e., a bounded interval with Lebesgue measure or a finite set with the counting measure). Regardless of our choice of algorithm, it is known that some such restriction is necessary in the worst case [ALMM19]. In most cases, the exact choice of $\mathcal{R}$ is not particularly critical for our algorithm, so we do not dwell on this issue.

There is a subtlety of our choice of loss function for the exponential mechanism: If $\hat{\theta}_i \neq \theta_i$ for all $\theta \in \Theta$, we have

$$
\max\left\{\left|\left\{\theta \in \Theta \; : \; \theta_i < \hat{\theta}_i\right\}\right|, \left|\left\{\theta \in \Theta \; : \; \theta_i > \hat{\theta}_i\right\}\right|\right\} =
$$
$$
\max\left\{\left|\left\{\theta \in \Theta \; : \; \theta_i < \hat{\theta}_i\right\}\right|, |\Theta| - \left|\left\{\theta \in \Theta \; : \; \theta_i < \hat{\theta}_i\right\}\right|\right\}
$$
$$
= \left|\left|\left\{\theta \in \Theta \; : \; \theta_i < \hat{\theta}_i\right\}\right| - \frac{1}{2}|\Theta|\right| + \frac{1}{2}|\Theta|.
$$

The final expression is more natural than the first expression. The quantity $\left|\left\{\theta \in \Theta \; : \; \theta_i < \hat{\theta}_i\right\}\right|$ gives the rank (i.e., rescaled quantile) of the value $\hat{\theta}_i$ in the multiset $\{\theta_i \; : \; \theta \in \Theta\}$. The true median has rank $\frac{1}{2}|\Theta|$, so the loss measures how far the rank is from this ideal. When everything has a continuous distribution, the above equality between the expressions holds with probability 1. However, if we have a discrete distribution (such as when $\mathcal{R}$ is a discrete set), the above equality does not hold. Consider the extreme case where the multiset $\Theta$ consists of a single point $\theta^*$ repeated many times. When $\hat{\theta}_i = \theta_i^*$, our loss function takes value 0 and, for $\hat{\theta}_i \neq \theta_i^*$, our loss function takes value $|\Theta|$. In contrast, the final expression above would yield a constant function taking value $|\Theta|$ everywhere. Thus our loss function performs better in the discrete case.

## 1.2 Our Results

We provide a theoretical privacy and utility analysis of our algorithm, as well as an experimental evaluation of our algorithm. Our theoretical guarantee is helpful to build understanding. However, our experimental results give a clearer comparison to prior work. See Figure 1 for an experimental comparison of algorithms. Next we state the main accuracy result:

**Theorem 1.1** (Main Result). *For any $\tilde{\varepsilon}, \tilde{\delta} > 0$ and $n, d, r \in \mathbb{N}$, Algorithm 1 with appropriate settings of parameters provides $(\tilde{\varepsilon}, \tilde{\delta})$-DP and the following accuracy guarantee.*

*Fix $\theta^* \in [-r, +r]^d$ and $\sigma > 0$. Assume the inputs $(x_1, y_1), (x_2, y_2), \ldots, (x_n, y_n) \in \mathbb{R}^d \times \mathbb{R}$ are drawn i.i.d. as follows. Independently for each $i \in [n]$, we have $x_i \leftarrow \mathcal{N}(0, I)$ and then, conditioned on $x_i$, we have $y_i \leftarrow \mathcal{N}(\langle \theta^*, x_i \rangle, \sigma^2)$.*

*If $\hat{\theta}$ is the output of Algorithm 1 with the above inputs and parameters, then, for all $\beta > 0$, we have*

$$
\mathbb{P}\left[\|\hat{\theta} - \theta^*\|_\infty \leq \sigma \cdot O\left(\frac{d \cdot \sqrt{d \cdot \log(1/\tilde{\delta})}}{\tilde{\varepsilon} n} \log\left(\frac{dr}{\beta}\right) + \sqrt{\frac{d \cdot \log(d/\beta)}{n}}\right) + \frac{1}{r}\right] \geq 1 - \beta.
$$

We now make some remarks about the meaning of our theoretical result.

**Pure DP vs. Approximate DP** Algorithm 1 offers both pure and approximate DP guarantees (and concentrated DP); see Proposition A.1 for details. The parameter $\varepsilon$ of the algorithm corresponds to the pure $(\varepsilon, 0)$-DP guarantee. In high dimensional settings (i.e., large $d$), we can apply advanced composition results to obtain better guarantees. Specifically, the approximate $(\tilde{\varepsilon}, \tilde{\delta})$-DP guarantee of Theorem 1.1 is achieved by setting $\varepsilon \approx \tilde{\varepsilon} \cdot \sqrt{\frac{d}{\log(1/\tilde{\delta})}}$.

**Accuracy Guarantee** The error bound of Theorem 1.1 has three terms: $\sigma \cdot \frac{d}{\varepsilon m} \log\left(\frac{dr}{\beta}\right)$ is the error due to privacy; $\sigma \cdot \sqrt{\frac{\log(d/\beta)}{m}}$ is the non-private statistical estimation error (a.k.a. generalization error); and $\frac{1}{r}$ is the error from rounding to the discrete set $\mathcal{R}$ of size $O(r^2)$.

Our accuracy guarantee bounds $\|\hat{\theta} - \theta^*\|_\infty$. This is particularly useful if our goal is to estimate some parameter $\theta_i^*$, as it provides a confidence interval. We can of course also use this to bound the Euclidean norm: $\|\hat{\theta} - \theta^*\|_2 \leq \sqrt{d} \cdot \|\hat{\theta} - \theta^*\|_\infty$. It is also common to provide bounds on the mean squared error. Under our distributional assumptions, this is equivalent to bounding the Euclidean norm: If $x \leftarrow \mathcal{N}(0, I)$ and $y \leftarrow \mathcal{N}(\langle \theta^*, x \rangle, \sigma^2)$, then, for all $\hat{\theta} \in \mathbb{R}^d$

$$\mathbb{E}\left[\left(\langle \hat{\theta}, x \rangle - y\right)^2\right] = \mathbb{E}\left[(\langle \theta^*, x \rangle - y)^2\right] + \|\hat{\theta} - \theta^*\|_2^2 = \sigma^2 + \|\hat{\theta} - \theta^*\|_2^2.$$

**Distributional Assumptions** We emphasize that our privacy guarantee is worst-case and the distributional assumptions are only for the accuracy analysis. Thus the maxim "all models are wrong, but some are useful" (attributed to George Box) applies. That is, we don't expect real data to perfectly follow a Gaussian distribution. Our algorithm still works even if these assumptions fail, but we believe that the theorem is a useful indication that our algorithm provides useful accuracy.

There is also some flexibility in the Gaussian assumption: If the $x_i$s are drawn from $\mathcal{N}(0, \Sigma)$ instead of $\mathcal{N}(0, I)$ then we can apply a transformation $(x_i, y_i) \mapsto \left(\Sigma^{-1/2} x_i, y_i\right)$ to make the distribution of $x_i$s spherical, run our algorithm to obtain $\hat{\theta}$, and then map this back to a solution to the original problem $\Sigma^{-1/2}\hat{\theta}$.

Our assumption that the data comes from a mulivariate Gaussian is reasonably standard. Assuming that $\|\theta^*\|_\infty \leq r$ is less standard. In the non-private setting we don't need to make any assumption on $\theta^*$, but it is necessary in the private case [ALMM19]. Note that we can arbitrarily rescale this constraint: If instead we assume $\|\theta^* - \theta^0\|_\infty \leq b \cdot r$ for some $b > 0$, then we can simply transform the data $(x_i, y_i) \mapsto \left(x_i, \frac{1}{b}(y_i - \langle \theta^0, x_i \rangle)\right)$, run our algorithm to obtain $\hat{\theta} \in [-r, r]^d$, and then map this back to a solution to the original problem $b \cdot \hat{\theta} + \theta^0$. The accuracy guarantee will be rescaled accordingly. Similarly, the infinity norm can be replaced by the Euclidean norm by transforming the problem with a random unitary matrix [e.g., KLS21, §4.2].

**Parameters** The sample size $n$, dimension $d$, noise variance $\sigma^2$, and privacy parameters $\tilde{\varepsilon}$ and $\tilde{\delta}$ are all standard parameters. The only non-standard parameter of Theorem 1.1 is $r$. This determines both the size and granularity of the restricted range $\mathcal{R}$ in Algorithm 1. This parameter should be thought of as capturing how uncertain we are about $\theta^* \in [-r, r]^d$ and how precise our final answer should be – i.e., the granularity of $\mathcal{R}$ is $1/r$ (which should ideally scale with $\sigma$). Theorem 1.1 runs Algorithm 1 with $\ell = 1$.

## 1.3 Related Work

Linear regression has been well studied in the non-private setting; we do not discuss this setting except to mention the connection to robust statistics. Robust statistics seeks to develop estimators that are resistant to a small fraction of the dataset being corrupted. This kind of robustness turns out to be useful for designing DP algorithms [NRS07; DL09; BS19] and our work extends this connection. In particular, the standard approach to linear regression is not robust, which led to the development of the robust Theil-Sen estimator [The50; Sen68] and its multivariate extension [DPWZ08], which are the basis for our work.

DP linear regression has also been well-studied. Most similar to our work is that of Alabi, McMillan, Sarathy, Smith, and Vadhan [AMSSV22], which studies the Theil-Sen estimator in the setting of *simple* linear regression. This is essentially our algorithm restricted to the case of $d = 1$, although they also add a constant intercept, i.e., an affine relationship $y \approx \theta x + b$. Adding an intercept is equivalent to adding an extra feature to $x$ that is always $1$ and adding a corresponding dimension to $\theta$. Dwork and Lei [DL09] propose two DP robust regression methods. The first is, like ours, based on the Theil-Sen estimator, although with a different method for computing the median. The second changes the loss function to one with bounded gradients, namely $\sum_i^n |\langle \theta, x_i \rangle - y_i|/\|x_i\|_2$, and

analyzes the robustness of the solution to this new problem. Unfortunately, Dwork and Lei [DL09] provide very limited theoretical results and no experimental results for us to compare against.

Our algorithmic approach of analyzing several subproblems and then privately combining the answers is based on the sample-and-aggregate framework of Nissim, Raskhodnikova, and Smith [NRS07]. Similar algorithms have appeared in other works. In particular, Feldman and Steinke [FS18] use a median-of-means approach to compute a univariate mean. Singhal and Steinke [SS21] propose an algorithm that is similar to ours, but for the different (but related) problem of finding a low-dimensional subspace that captures the data.

A natural approach to DP linear regression is to apply general-purpose optimization tools to the objective function $f(\theta) = \sum_i^n (\langle \theta, x_i \rangle - y_i)^2$. Noisy gradient descent (DP-GD) [SCS13; BST14; ACGMMTZ16] is a widely-used tool for private optimization. It adds noise to the gradients $\nabla f(\theta) = 2 \sum_i^n (\langle \theta, x_i \rangle - y) \cdot x_i$ encountered during the optimization procedure. To ensure that the gradients are bounded, we must clip them before addoing noise. That is, we add noise to $\min\{1, c/\|\nabla f(\theta)\|\} \cdot \nabla f(\theta)$ instead of $\nabla f(\theta)$, which could be unbounded. This approach works remarkably well, but it requires carefully setting the clipping parameter $c$. The larger $c$ is, the more noise we add. But if $c$ is too small we distort the gradients and the optimization procedure may not even converge in time. We use this approach as a comparison point in our experiments, but we find that setting the parameters ($c$, number of steps, and learning rate) to be highly non-trivial. In an unpublished work, Varshney, Jain, and Thakurta [VJT22] propose a variant of DP-GD where the clipping parameter $c$ is chosen in a data-dependent manner at each step of the optimization. They show that this adaptive clipping can achieve asymptotically optimal results. Kamath, Li, Singhal, and Ullman [KLSU19] apply a similar adaptive clipping approach to learning the parameters of a Gaussian distribution; linear regression can be reduced to this task [MKFI22]. Another general-purpose optimization tool is Objective Perturbation [CMS11], which was applied to linear regression by Wang [Wan18], but objective perturbation requires stronger assumptions than DP-GD (such as convexity ans smoothness) which means we also need additional assumptions to apply it to linear regression. Finally, we mention that, under the right assumptions, it is possible to apply the exponential mechanism [MT07b] to the linear regression objective, which can be viewed as a form of bayesian sampling [Wan18].

Since there is a closed-form solution in the non-private setting – namely, $\hat{\theta} = (X^T X)^{-1} X^T y$ where each example $(x_i, y_i)$ is a row of $X$ and the corresponding row of $y$ – another natural approach to the problem is to perturb $X^T X = \sum_i^n x_i x_i^T \in \mathbb{R}^{d \times d}$ and $X^T y = \sum_i^n y_i x_i \in \mathbb{R}^d$, which are known as the sufficient statistics. This requires us to bound the sensitivity of these terms, which boils down to bounding $\|x_i\|_2$ and $|y_i|$. For our experimental comparison, we add Gaussian noise to both $X^T X$ and $X^T y$. One downside of adding Gaussian noise to $X^T X$ is that it may cease to be positive semidefinite. Thus it has also been suggested to add noise drawn from a Wishart distribution [She19]. (We note that analyzing Wishart noise is difficult and incorrect analyses of this approach have been published [JXZ16; IS16].) Wang [Wan18] also studied an adaptive form of sufficient statistics perturbation.

It is also possible to add noise directly to the data [DTTZ14; She17; She19]. That is, we perturb $X$ and $y$, which also requires bounding $\|x_i\|_2$ and $|y_i|$. This tends to yield worse results than perturbing the sufficient statistics. Intuitively, this approach adds noise to each of the $n$ rows of $X$ and $y$, so the amount of noise grows with $n$. In contrast, the amount of noise added to $X^T X$ and $X^T y$ does not grow with $n$. However, adding noise to the data is desirable if we are in the setting of local DP [KLNRS11]; our results are for the central DP setting.

As mentioned earlier, of the key advantages of our algorithm over the optimization and perturbation approaches is that we do not need to clip or bound the data $(x_i, y_i)$, which can be quite detrimental to accuracy in practice. Our use of a median-based algorithm means we have much lower sensitivity to these bounds (logarithmic instead of linear).

# 2   Experiments

We now perform an empirical evaluation of our algorithm using synthetic data. We compare to state-of-the-art approaches and, since our algorithm has several moving parts, we also consider variants of our algorithm.

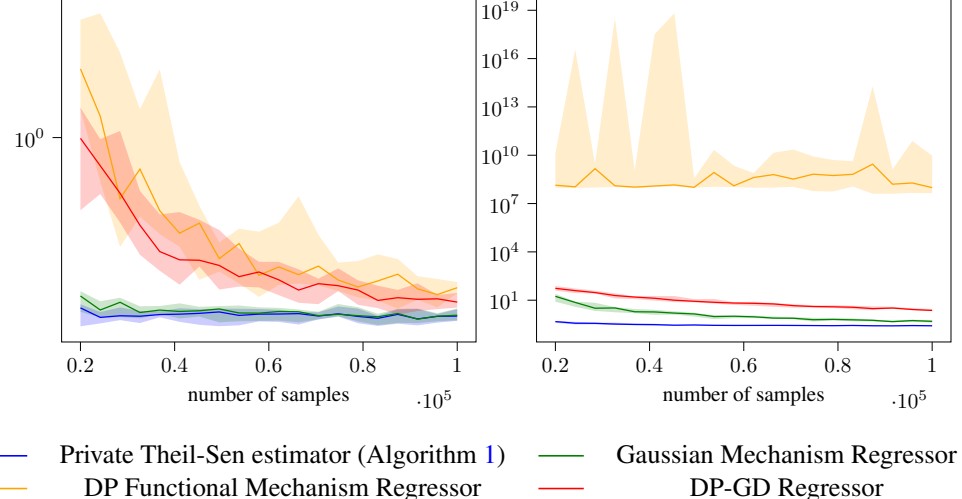

Figure 2: Comparison of DP linear regression algorithms for features sampled from $\mathcal{N}(0, I)$. Mean square error (i.e., $\mathbb{E}[(\langle\hat{\theta}, x\rangle - y)^2]$ on vertical axis in logarithmic scale) as a function of the number of samples (i.e., $n$ on horizontal axis) for dimensions $d = 10$ (left) and $d = 30$ (right); and a function of $\varepsilon$ for $n = 10^5$ and dimensions $d = 10$ (left) and $d = 30$ (right). The line show the median and the semitransparent shadow shows the 0.1 and 0.9 quantiles of the error; values are computed over 20 runs. Privacy parameters are $\varepsilon = 1$ and $\delta = 10^{-6}$; and $\ell = 1$. Data is synthetic, see Section 2.1 for details.

## 2.1 Synthetic Data

We perform our experiments using synthetic data, as this allows us to be precise about what assumptions we are and are not making. In all these experiments $\theta$ is sampled uniformly from $[-1, 1]^d$, features $x_1, \ldots, x_n$ are sampled independently and uniformly from $[0, 1]^d$ and each $y_i$ is sampled from $\mathcal{N}(\langle\theta, x_i\rangle, \sigma^2)$ independently (conditioned on $x_i$), where $\sigma = 0.1$.

Note that the features are sampled from a bounded distribution, rather than a Gaussian as in Theorem 1.1. *We make this choice in order to be generous to the algorithms we compare against.* The algorithms we compare against clip the data or gradients before adding noise, so we make the problem easier for them by ensuring that the data is in fact bounded – i.e., we ensure that the clipping does not distort the data. Our algorithm does not require this kind of assumption on the features: Figure 2 shows the errors if the features are sampled from $\mathcal{N}(0, I)$.

## 2.2 Private Algorithms

We run Algorithm 1 with $\ell = 1$ and $\mathcal{R} = [-1, 1]$. For comparison, we run the following state-of-the-art regression algorithms:

- **DP-GD based regressor:** This algorithm applies noisy gradient descent to minimize the loss $\sum_{i=1} \left(\left\langle\hat{\theta}, x_i\right\rangle - y_i\right)^2$. The learning rate is 0.1, the number of epochs is 100, and the clipping rate is $8d$. (Our implementation of private GD gives result similar to the results obtained by running DP-SGD provided by TensorFlow Privacy.)

- **Gaussian covariate matrix perturbation regressor:** This algorithm outputs $\hat{\theta} = (X^T X + A)^{-1}(X^T y + b)$, where $A$ is an appropriately scaled Gaussian matrix of size $d \times d$ and $b$ is a Gaussian vector of size $d$.

- **Functional mechanism based regressor:** This algorithm represents the loss function $\sum_i^n \left(\left\langle\hat{\theta}, x_i\right\rangle - y_i\right)^2$ as a polynomial in $\hat{\theta}_1, \ldots, \hat{\theta}_d$ add appropriately scaled Laplacian noise to each coefficient of the polynomial to obtain $\hat{p}$ and uses the Broy-

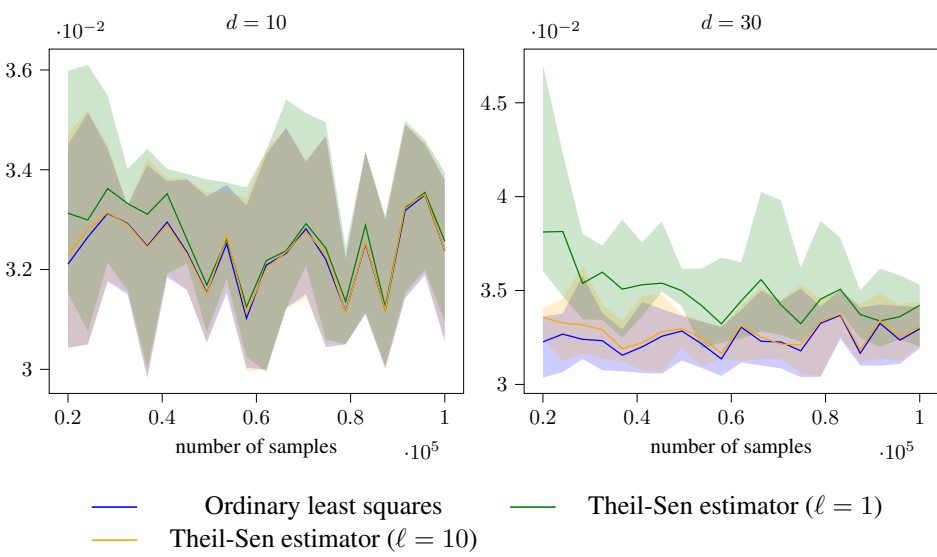

Figure 3: Mean square error as a function of the number of samples for $d = 10$ and $d = 30$. The semitransparent shadow shows values between $0.1$ and $0.9$ quantiles of the accuracy.

den–Fletcher–Goldfarb–Shanno algorithm to find $\hat{\theta}$ minimizing $\hat{p}$; we use the implementation provided by Holohan et al. [HBMAL19].

Figure 1 shows that the error of our algorithm is lower that of the other algorithms we compare against.

## 2.3 Non-private Algorithms

Before analyzing performance of private algorithms let us study performance on the non-private version of the private efficient Theil-Sen estimator: non-private can be obtained from Algorithm 1 by replacing Line 15 by a line that sets $\hat{\theta}_i$ such that $\max\left\{\left|\left\{\theta \in \Theta : \theta_i < \hat{\theta}_i\right\}\right|, \left|\left\{\theta \in \Theta : \theta_i > \hat{\theta}_i\right\}\right|\right\} = \frac{m}{2}$.

Figure 3 shows that for reasonably large values of $\ell$, efficient multivariate Theil-Sen estimator performs as well as ordinary least squares estimator.

## 2.4 Values of $\ell$

This section analyses relative performance of $\ell$-partition DP Theil–Sen for different values of $\ell$: we considered $\ell \in \{1, 10, 20\}$. Figure 4 shows that their convergence rates are comparable in contrast with the non private setting where increasing $\ell$ improves the accuracy: this effect can be explained by the fact that the median heuristic uses amount of budget proportional to $1/\ell$ so increasing $\ell$ improves the true median, but adds more noise.

Because of this observation we only consider $\ell = 1$.

## 2.5 Algorithms for Median

This section is analysing relative performance of efficient private Theil–Sen estimator for two choices of differentially private median heuristics: private median based on exponential mechanism that is used in Algorithm 1 and private median based on widened exponential mechanism defined in [AMSSV22]. Figure 5 shows that like in case of $d = 1$ [AMSSV22], private median based on exponential mechanism performs better on synthetic data.

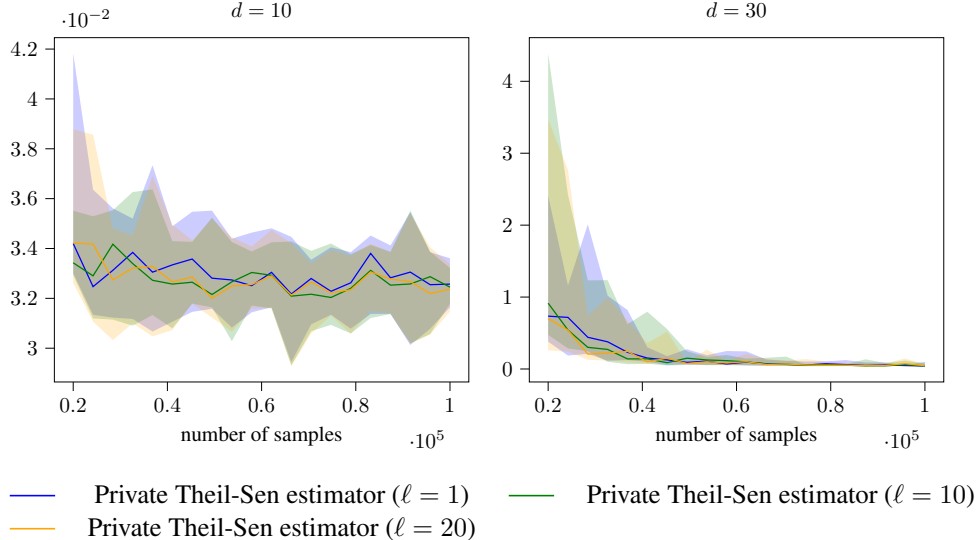

Figure 4: Mean square error as a function of the number of samples for $d = 10$ and $d = 30$. The semitransparent shadow shows values between $0.1$ and $0.9$ quantiles of the error.

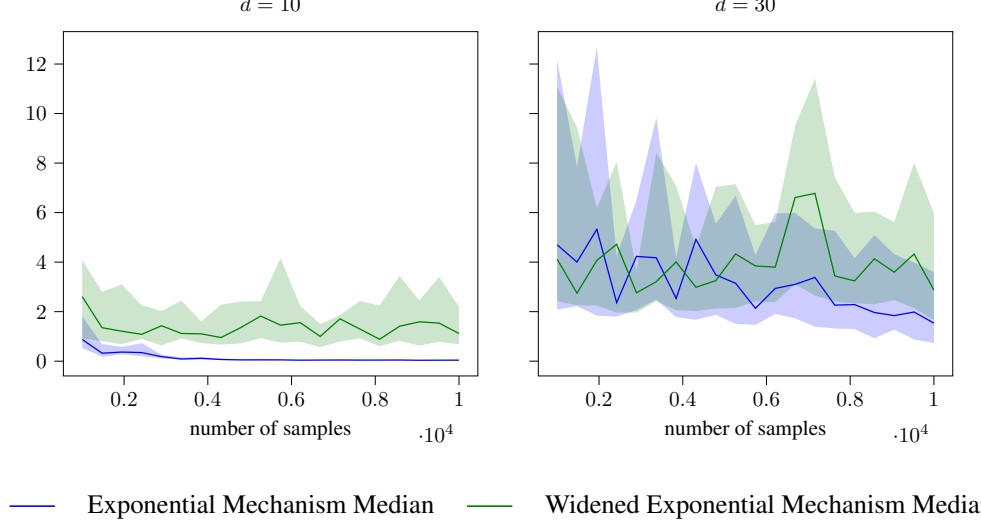

Figure 5: Mean square error as a function of the number of samples for $d = 10$ and $d = 30$. The semitransparent shadow shows values between $0.1$ and $0.9$ quantiles of the error.

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

| | |
|---|---|
| [DR14] | C. Dwork and A. Roth. "The algorithmic foundations of differential privacy." In: *Found. Trends Theor. Comput. Sci.* 9.3-4 (2014), pp. 211–407. URL: https://www.cis.upenn.edu/~aaroth/Papers/privacybook.pdf (cit. on p. 14). |
| [DR16] | C. Dwork and G. N. Rothblum. "Concentrated differential privacy". In: *arXiv preprint arXiv:1603.01887* (2016) (cit. on p. 3). |
| [DTTZ14] | C. Dwork, K. Talwar, A. Thakurta, and L. Zhang. "Analyze gauss: optimal bounds for privacy-preserving principal component analysis". In: *Proceedings of the forty-sixth annual ACM symposium on Theory of computing*. 2014, pp. 11–20 (cit. on p. 6). |
| [FS18] | V. Feldman and T. Steinke. "Calibrating noise to variance in adaptive data analysis". In: *Conference On Learning Theory*. PMLR. 2018, pp. 535–544 (cit. on pp. 3, 6). |
| [GJK21] | J. Gillenwater, M. Joseph, and A. Kulesza. "Differentially private quantiles". In: *International Conference on Machine Learning*. PMLR. 2021, pp. 3713–3722 (cit. on p. 3). |
| [HBMAL19] | N. Holohan, S. Braghin, P. Mac Aonghusa, and K. Levacher. "Diffprivlib: the IBM differential privacy library". In: *ArXiv e-prints* 1907.02444 [cs.CR] (July 2019) (cit. on p. 8). |
| [IS16] | H. Imtiaz and A. D. Sarwate. "Symmetric matrix perturbation for differentially-private principal component analysis". In: *2016 IEEE International Conference on Acoustics, Speech and Signal Processing (ICASSP)*. IEEE. 2016, pp. 2339–2343 (cit. on p. 6). |
| [JXZ16] | W. Jiang, C. Xie, and Z. Zhang. "Wishart mechanism for differentially private principal components analysis". In: *Proceedings of the AAAI Conference on Artificial Intelligence*. Vol. 30. 1. 2016 (cit. on p. 6). |
| [KLMNS20] | H. Kaplan, K. Ligett, Y. Mansour, M. Naor, and U. Stemmer. "Privately learning thresholds: Closing the exponential gap". In: *Conference on Learning Theory*. PMLR. 2020, pp. 2263–2285 (cit. on p. 3). |
| [KLNRS11] | S. P. Kasiviswanathan, H. K. Lee, K. Nissim, S. Raskhodnikova, and A. Smith. "What can we learn privately?" In: *SIAM Journal on Computing* 40.3 (2011), pp. 793–826 (cit. on p. 6). |
| [KLS21] | P. Kairouz, Z. Liu, and T. Steinke. "The distributed discrete gaussian mechanism for federated learning with secure aggregation". In: *International Conference on Machine Learning*. PMLR. 2021, pp. 5201–5212 (cit. on p. 5). |
| [KLSU19] | G. Kamath, J. Li, V. Singhal, and J. Ullman. "Privately learning high-dimensional distributions". In: *Conference on Learning Theory*. PMLR. 2019, pp. 1853–1902 (cit. on pp. 3, 6). |
| [KSU20] | G. Kamath, V. Singhal, and J. Ullman. "Private mean estimation of heavy-tailed distributions". In: *Conference on Learning Theory*. PMLR. 2020, pp. 2204–2235 (cit. on p. 1). |
| [KV17] | V. Karwa and S. Vadhan. "Finite sample differentially private confidence intervals". In: *arXiv preprint arXiv:1711.03908* (2017) (cit. on pp. 1, 3). |
| [LKKO21] | X. Liu, W. Kong, S. Kakade, and S. Oh. "Robust and differentially private mean estimation". In: *Advances in Neural Information Processing Systems*. Ed. by M. Ranzato, A. Beygelzimer, Y. Dauphin, P. Liang, and J. W. Vaughan. Vol. 34. Curran Associates, Inc., 2021, pp. 3887–3901. URL: https://proceedings.neurips.cc/paper/2021/file/1fc5309ccc651bf6b5d22470f67561ea-Paper.pdf (cit. on p. 1). |
| [LKO21] | X. Liu, W. Kong, and S. Oh. "Differential privacy and robust statistics in high dimensions". In: *arXiv preprint arXiv:2111.06578* (2021) (cit. on p. 1). |
| [Mas90] | P. Massart. "The tight constant in the Dvoretzky-Kiefer-Wolfowitz inequality". In: *The annals of Probability* (1990), pp. 1269–1283 (cit. on p. 15). |

[MKFI22]   J. Milionis, A. Kalavasis, D. Fotakis, and S. Ioannidis. "Differentially Private Regression with Unbounded Covariates". In: *Proceedings of The 25th International Conference on Artificial Intelligence and Statistics*. Ed. by G. Camps-Valls, F. J. R. Ruiz, and I. Valera. Vol. 151. Proceedings of Machine Learning Research. PMLR, Mar. 2022, pp. 3242–3273. URL: https://proceedings.mlr.press/v151/milionis22a.html (cit. on p. 6).

[MT07a]   F. McSherry and K. Talwar. "Mechanism design via differential privacy". In: *48th Annual IEEE Symposium on Foundations of Computer Science (FOCS'07)*. IEEE. 2007, pp. 94–103 (cit. on p. 3).

[MT07b]   F. McSherry and K. Talwar. "Mechanism Design via Differential Privacy". In: *48th Annual IEEE Symposium on Foundations of Computer Science (FOCS 2007), October 20-23, 2007, Providence, RI, USA, Proceedings*. IEEE Computer Society, 2007, pp. 94–103. URL: https://doi.org/10.1109/FOCS.2007.41 (cit. on p. 6).

[MT22]   B. McMahan and A. Thakurta. *Federated Learning with Formal Differential Privacy Guarantees*. https://ai.googleblog.com/2022/02/federated-learning-with-formal.html. 2022 (cit. on p. 1).

[NRS07]   K. Nissim, S. Raskhodnikova, and A. Smith. "Smooth sensitivity and sampling in private data analysis". In: *Proceedings of the thirty-ninth annual ACM symposium on Theory of computing*. 2007, pp. 75–84 (cit. on pp. 3, 5, 6).

[RS21]   R. Rogers and T. Steinke. *A Better Privacy Analysis of the Exponential Mechanism*. DifferentialPrivacy.org. https://differentialprivacy.org/exponential-mechanism-bounded-range/. July 2021 (cit. on p. 14).

[SCS13]   S. Song, K. Chaudhuri, and A. D. Sarwate. "Stochastic gradient descent with differentially private updates". In: *2013 IEEE Global Conference on Signal and Information Processing*. IEEE. 2013, pp. 245–248 (cit. on p. 6).

[Sen68]   P. K. Sen. "Estimates of the regression coefficient based on Kendall's tau". In: *Journal of the American statistical association* 63.324 (1968), pp. 1379–1389 (cit. on pp. 2, 5).

[She17]   O. Sheffet. "Differentially Private Ordinary Least Squares". In: *Proceedings of the 34th International Conference on Machine Learning*. Ed. by D. Precup and Y. W. Teh. Vol. 70. Proceedings of Machine Learning Research. PMLR, Aug. 2017, pp. 3105–3114. URL: https://proceedings.mlr.press/v70/sheffet17a.html (cit. on p. 6).

[She19]   O. Sheffet. "Old techniques in differentially private linear regression". In: *Algorithmic Learning Theory*. PMLR. 2019, pp. 789–827 (cit. on p. 6).

[Smi08]   A. Smith. "Efficient, differentially private point estimators". In: *arXiv preprint arXiv:0809.4794* (2008) (cit. on p. 3).

[Smi11]   A. D. Smith. "Privacy-preserving statistical estimation with optimal convergence rates". In: *Proceedings of the 43rd ACM Symposium on Theory of Computing, STOC 2011, San Jose, CA, USA, 6-8 June 2011*. Ed. by L. Fortnow and S. P. Vadhan. ACM, 2011, pp. 813–822. URL: https://doi.org/10.1145/1993636.1993743 (cit. on p. 3).

[SS21]   V. Singhal and T. Steinke. "Privately learning subspaces". In: *Advances in Neural Information Processing Systems* 34 (2021) (cit. on p. 6).

[Sza91]   S. J. Szarek. "Condition numbers of random matrices". In: *Journal of Complexity* 7.2 (1991), pp. 131–149. ISSN: 0885-064X. URL: https://www.sciencedirect.com/science/article/pii/0885064X9190002F (cit. on p. 15).

[The50]   H. Theil. "A rank-invariant method of linear and polynomial regression analysis". In: *Indagationes mathematicae* 12.85 (1950), p. 173 (cit. on pp. 2, 5).

[VJT22]   P. Varshney, P. Jain, and A. Thakurta. *(Nearly) Optimal Private Linear Regression via Adaptive Clipping*. (personal communication). 2022 (cit. on p. 6).

[Wan18]   Y.-X. Wang. "Revisiting differentially private linear regression: optimal and adaptive prediction & estimation in unbounded domain". In: *arXiv preprint arXiv:1803.02596* (2018) (cit. on p. 6).


# A   Proof of Main Result

In this section, we prove Theorem 1.1. The proof is split into two parts: Proposition A.1 analyzes the privacy (which is a worst case property). Proposition A.2 analyzes the accuracy (which requires distributional assumptions).

**Proposition A.1** (Privacy). *Algorithm 1 satisfies $\varepsilon$-DP and also $\frac{\varepsilon^2}{8d}$-zCDP [BS16].*

Our privacy proof follows the standard template of using the properties of the exponential mechanism along with the composition property of differential privacy.

In addition to a pure DP guarantee, we provide a concentrated DP guarantee. In high dimensional settings, concentrated DP is preferable. To achieve $\rho$-zCDP, we would set $\varepsilon = \sqrt{8d\rho}$. Note that $\rho$-zCDP can be converted to $(\varepsilon, \delta)$-DP for any $\varepsilon \geq 0$ and $\delta = \inf_{\alpha>1} e^{(\alpha-1)(\alpha\rho-\varepsilon)} \cdot \left(1 - \frac{1}{\alpha}\right)^{\alpha-1} \cdot \frac{1}{\alpha}$ [e.g., CKS20, Cor. 13]. Asymptotically, to achieve approximate $(\tilde{\varepsilon}, \tilde{\delta})$-DP it suffices to set $\rho =$

$\Theta\left(\frac{\tilde{\varepsilon}^2}{\log(1/\tilde{\delta})}\right)$ [BS16, Lem. 3.5]. The privacy claim of Theorem 1.1 follows by substituting $\varepsilon = \sqrt{8d\rho} = \Theta(\tilde{\varepsilon} \cdot \sqrt{d/\log(1/\tilde{\delta})})$ into Proposition A.1.

*Proof of Proposition A.1.* Algorithm 1 invokes the exponential mechanism $d$ times. We analyze one invocation and then apply composition.

The loss function $\max\left\{\left|\left\{\theta \in \Theta : \theta_i < \hat{\theta}_i\right\}\right|, \left|\left\{\theta \in \Theta : \theta_i > \hat{\theta}_i\right\}\right|\right\}$ has sensitivity 1 in terms of changing an element of the multiset $\Theta$. This is because it is the maximum of two counts. Each count naturally has sensitivity 1 and the maximum does not increase the sensitivity. Changing one input $(x_i, y_i)$ can change $\ell$ elements of $\Theta$, as that input may appear in up to $\ell$ subproblems $S_{j,k}$. Thus the loss function has sensitivity $\ell$ in terms of changing one input.

The distribution we sample from is

$$\mathbb{P}[\hat{\theta}_i] \propto \exp\left(-\frac{\varepsilon}{2\ell d} \max\left\{\left|\left\{\theta \in \Theta : \theta_i < \hat{\theta}_i\right\}\right|, \left|\left\{\theta \in \Theta : \theta_i > \hat{\theta}_i\right\}\right|\right\}\right).$$

Note that the multiplier $\frac{\varepsilon}{2\ell d}$ is $\varepsilon/d$ divided by twice the sensitivity. Thus [DR14, Thm. 3.10] tells us that this sampling procedure is $(\varepsilon/d, 0)$-DP. Since we invoke this exponential mechanism independently $d$ times (for all the coordinates of $\hat{\theta}$), we can apply basic composition [DR14, Thm. 3.14] to show that the overall algorithm is $(\varepsilon, 0)$-DP.

For the concentrated DP analysis, we can use an improved analysis of the exponential mechanism [RS21] that show that, in addition to $(\varepsilon/d, 0)$-DP, each invocation of the exponential mechanism satisfies $\frac{1}{8}(\varepsilon/d)^2$-zCDP. Finally, we can apply composition for concentrated DP [BS16] over the $d$ invocations to show that the overall algorithm is $\rho$-zCDP with $\rho = \frac{1}{8}(\varepsilon/d)^2 \cdot d = \varepsilon^2/8d$. $\qquad\square$

Next we provide a theoretical utility guarantee. However, the proof of the pudding is in the eating, so we direct the reader to the experimental results in Section 2 to see how our algorithm performs in practice.

**Proposition A.2** (Accuracy). *Fix the parameters $\varepsilon > 0$, $n, d, r \in \mathbb{N}$, $\ell = 1$, $m = \lfloor n/d \rfloor$, and $\mathcal{R} = \left\{-1 + 2\frac{i-1}{r-1} : i \in [r]\right\}$ of Algorithm 1. Let us also fix $\theta^* \in [-1, +1]^d$ and $\sigma > 0$. Assume the inputs $(x_1, y_1), (x_2, y_2), \ldots, (x_n, y_n) \in \mathbb{R}^d \times \mathbb{R}$ are drawn i.i.d. as follows. Independently for each $i \in [n]$, we have $x_i \leftarrow \mathcal{N}(0, I)$ and then, conditioned on $x_i$, we have $y_i \leftarrow \mathcal{N}(\langle \theta^*, x_i \rangle, \sigma^2)$. If $\hat{\theta}$ is the output of Algorithm 1 with the above inputs and parameters, then, for all $\beta > 0$, we have*

$$\mathbb{P}\left[\|\hat{\theta} - \theta^*\|_\infty \leq \frac{1}{r-1} + \sigma \cdot O\left(\frac{d}{\varepsilon m}\log\left(\frac{dr}{\beta}\right) + \sqrt{\frac{\log(d/\beta)}{m}}\right)\right] \geq 1 - \beta,$$

*where the probability is over both the randomness of the algorithm and the inputs.*

The proof consists of three steps: First, we use the properties of the exponential mechanism to show that Algorithm 1 outputs a point with low empirical loss. Second, we use a generalization result to show that the output also has low population loss. Third, we use the distributional assumptions in the theorem to show that low population loss implies that the output is indeed close to the desired value.

The first lemma shows that, with high probability, the empirical loss is low.

**Lemma A.3.** *Let $\varepsilon$, $\ell$, $d$, $m$, $\mathcal{R}$, $\Theta$, and $\hat{\theta}$ be as in Algorithm 1. Assume $|\mathcal{R}| < \infty$. Independently, for each $i$ and all $\beta > 0$, we have*

$$\mathbb{P}_{\hat{\theta}_i}\left[\max\left\{\left|\left\{\theta \in \Theta : \theta_i < \hat{\theta}_i\right\}\right|, \left|\left\{\theta \in \Theta : \theta_i > \hat{\theta}_i\right\}\right|\right\} \leq \left\lfloor\frac{|\Theta|}{2}\right\rfloor + \frac{2\ell d}{\varepsilon}\log\left(\frac{|\mathcal{R}|}{\beta}\right)\right] \geq 1 - \beta.$$

*Proof.* First, we note that the empirical median of $\{\theta_i : \theta \in \Theta\}$ has loss at most $\lfloor|\Theta|/2\rfloor$. Combining this with the standard utility analysis of the exponential mechanism [DR14, Thm. 3.11] yields the result. Finally, we remark that Algorithm 1 samples each coordinate of $\hat{\theta}$ independently. $\qquad\square$

Our second lemma helps us relate the empirical loss to the population loss. That is, it is a generalization result.

**Lemma A.4** (DKW inequality [DKW56; Mas90])**.** *There exists a universal finite constant $C > 0$ such that the following holds. Let $F$ be the cumulative distribution function (CDF) of a probability distribution on $\mathbb{R}$ and let $X_1, X_2, \ldots, X_m$ be independent samples from that distribution – i.e., $F(x) = \mathbb{P}[X_i \leq x]$ for all $i \in [m]$. Then*

$$\forall \beta > 0 \quad \mathbb{P}_X \left[ \sup_{x \in \mathbb{R}} \left| \frac{1}{m} \sum_i^m \mathbb{I}[X_i \leq x] - F(x) \right| \leq \sqrt{\frac{\log(C/\beta)}{2m}} \right] \geq 1 - \beta.$$

Now we bring the distributional assumptions of Proposition A.2 into the analysis. For each of the subproblems, we are given $X \in \mathbb{R}^{d \times d}$ and $y \in \mathbb{R}^d$. The assumption is that $X$ consists of i.i.d. standard Gaussian entries and that $y = X\theta^* + z$, where $z \leftarrow \mathcal{N}(0, \sigma^2 I)$ is noise. Our goal is to estimate the unknown true parameters $\theta^* \in \mathbb{R}^d$. The estimate for the subproblem is $\theta = X^{-1}y = \theta^* + X^{-1}z$. Thus we need to understand the distribution of $X^{-1}z$. We begin by bounding the norm of $X^{-1}$. Then, in Lemma A.6, we use this bound to show that the distribution of $X^{-1}z$ is sufficiently concentrated around zero, which is necessary to ensure the median is well-behaved.

**Lemma A.5.** *Let $X \in \mathbb{R}^{d \times d}$ have entries which are all independent standard Gaussians. There exists a universal constant $C$ such that for all $\gamma \in (0, 3/4)$,*

$$\mathbb{P}\left[ \|X^{-1}\|_F^2 = \text{trace}((XX^T)^{-1}) \leq C \cdot d/\gamma^2 \right] \geq 1 - \gamma.$$

*Proof.* Let $\lambda_1(XX^T) \leq \lambda_2(XX^T) \leq \cdots \leq \lambda_d(XX^T)$ denote the eigenvalues of $XX^T$ in sorted order with multiplicities. Then $\text{trace}((XX^T)^{-1}) = \sum_j^d 1/\lambda_j(XX^T)$.

Szarek [Sza91] (Thm. 1.2, eq. 1.2) shows that, for all $j \in [d]$ and all $\alpha \geq 0$,

$$\mathbb{P}\left[ \sqrt{\lambda_j(XX^T)} < \frac{\alpha \cdot j}{\sqrt{d}} \right] \leq (\sqrt{2e} \cdot \alpha)^{j^2}.$$

By a union bound, for all $\alpha \in [0, 1/4]$, we have

$$\mathbb{P}\left[ \forall j \in [d] \ \lambda_j(XX^T) \geq \frac{\alpha^2 \cdot j^2}{d} \right] \geq 1 - \sum_{j=1}^d (\sqrt{2e} \cdot \alpha)^{j^2} \geq$$

$$1 - \sum_{j=1}^\infty (\sqrt{2e} \cdot \alpha)^{1+3(j-1)} = 1 - \frac{\sqrt{2e} \cdot \alpha}{1 - (\sqrt{2e} \cdot \alpha)^3} \geq 1 - 3\alpha.$$

If $\lambda_j(XX^T) \geq \frac{\alpha^2 \cdot j^2}{n}$ for all $j \in [d]$, then

$$\text{trace}((XX^T)^{-1}) = \sum_j^d 1/\lambda_j(XX^T) \leq \sum_{j=1}^\infty \frac{d}{\alpha^2 \cdot j^2} = \frac{\pi^2 d}{6\alpha^2}.$$

To complete the proof, set $\alpha = \gamma/3 \leq 1/4$ and $C = 3\pi^2/2 < 15$ □

**Lemma A.6.** *Let $X \in \mathbb{R}^{d \times d}$ and $y \in \mathbb{R}^d$ have entries which are all independent standard Gaussians. Let $u \in \mathbb{R}^d$ be an arbitrary unit vector that is independent from $X$ and $y$. Then the distribution of $u^T X^{-1} y$ is continuous and symmetric around $0$. Furthermore, for all $t \geq 0$,*

$$\mathbb{P}\left[ |u^T X^{-1} y| \leq t \right] \geq \frac{1}{2}\mathbb{P}\left[ |g| \leq \frac{t}{8\sqrt{C}} \right],$$

*where $g$ is a standard Gaussian and $C$ is the univeral constant from Lemma A.5.*

*Proof.* The distribution of $u^T X^{-1} y$ is a mixture of centered univariate Gaussians. Specifically, if $u$ and $X$ are fixed, then $u^T X^{-1} y \sim \mathcal{N}(0, \|(u^T X^{-1})^T\|_2^2)$. The randomness of $u$ and $X$ induces a

533 mixture. From this it is immediate that the distribution is symmetric about $0$ and that it is continuous
534 (as $\mathbb{P}[\|(u^T X^{-1})^T\|_2 = 0] = 0$).

535 Since the distribution of $X$ is spherically symmetric, so too is that of $X^{-1}$. This means that the
536 choice of $u$ is irrelevant. In particular, we can assume that $u$ is a uniformly random unit vector
537 (independent from everything else).

538 Fix $t \geq 0$ and $s \geq 0$. Let $g$ denote a standard univariate Gaussian (independent from everything
539 else). Then

$$\mathbb{P}[|u^T X^{-1} y| \leq t] = \mathop{\mathbb{E}}_{u,X} \left[ \mathbb{P}_y[|u^T X^{-1} y| \leq t] \right]$$

$$= \mathop{\mathbb{E}}_{u,X} \left[ \mathbb{P}_g[\|(u^T X^{-1})^T\|_2 \cdot |g| \leq t] \right]$$

$$\geq \mathbb{P}_g[|g| \leq t/s] \cdot \mathbb{P}_{u,X}[\|(u^T X^{-1})^T\|_2 \leq s].$$

540 Now we need to bound $\|(u^T X^{-1})^T\|_2$. We can express this quantity in terms of the Frobenius
541 matrix inner product:

$$\|(u^T X^{-1})^T\|_2^2 = u^T (XX^T)^{-1} u = \left\langle (XX^T)^{-1}, uu^T \right\rangle.$$

542 We have $\mathbb{E}\left[uu^T\right] = \frac{1}{d} I.$ [2] Thus, by linearity of expectation,

$$\mathop{\mathbb{E}}_u \left[\|(u^T X^{-1})^T\|_2^2\right] = \left\langle (XX^T)^{-1}, \frac{1}{d} I \right\rangle = \frac{1}{d} \text{trace}((XX^T)^{-1}) = \frac{1}{d}\|X^{-1}\|_F^2.$$

543 Fix $v \geq 1/\sqrt{d}$. By Markov's inequality,

$$\mathbb{P}_u \left[\|(u^T X^{-1})^T\|_2 \leq v\|X^{-1}\|_F\right] =$$

$$1 - \mathbb{P}_u \left[\|(u^T X^{-1})^T\|_2^2 > v^2\|X^{-1}\|_F^2\right] \geq$$

$$1 - \frac{\mathop{\mathbb{E}}_u \left[\|(u^T X^{-1})^T\|_2^2\right]}{v^2\|X^{-1}\|_F^2} = 1 - \frac{1}{dv^2}.$$

544 Thus

$$\mathbb{P}_{u,X} \left[\|(u^T X^{-1})^T\|_2 \leq s\right] \geq$$

$$\mathbb{P}_u \left[\|(u^T X^{-1})^T\|_2 \leq v\|X^{-1}\|_F\right] \cdot \mathbb{P}_X \left[\|X^{-1}\|_F \leq s/v\right] \geq$$

$$\left(1 - \frac{1}{dv^2}\right) \cdot \mathbb{P}_X \left[\|X^{-1}\|_F \leq s/v\right].$$

545 Lemma A.5 bounds $\|X^{-1}\|_F$, namely $\mathbb{P}_X[\|X^{-1}\|_F \geq s/v] \geq 1 - \sqrt{Cd} \cdot v/s \geq 1/4$ for some
546 universal constant $C$. Putting everything together gives

$$\mathbb{P}\left[|u^T X^{-1} y| \leq t\right] \geq \mathbb{P}_g\left[|g| \leq t/s\right] \cdot \left(1 - \frac{1}{dv^2}\right) \cdot \left(1 - \frac{\sqrt{Cd}v}{s}\right).$$

547 We set $v = 2/\sqrt{d}$ and $s = 8\sqrt{C}$, which gives

$$\mathbb{P}\left[|u^T X^{-1} y| \leq t\right] \geq \mathbb{P}_g\left[|g| \leq t/8\sqrt{C}\right] \cdot \frac{3}{4} \cdot \frac{3}{4}.$$

548 $\qquad\qquad\qquad\qquad\qquad\qquad\qquad\qquad\qquad\qquad\qquad\qquad\qquad\qquad\qquad\qquad\qquad$ $\square$

549 Now it is time to assemble the proof:

---

[2] To see this, imagine $u = u_1$ is generated by taking a column of a uniformly random unitary matrix $U = (u_1, u_2, \ldots, u_d) \in \mathbb{R}^{d \times d}$. By symmetry, $\mathbb{E}\left[u_1 u_1^T\right] = \mathbb{E}\left[u_2 u_2^T\right] = \cdots = \mathbb{E}\left[u_d u_d^T\right]$. Since $\sum_i^d u_i u_i^T = UU^T = I$, we have $\mathbb{E}\left[u_1 u_1^T\right] = \frac{1}{d} I$.

*Proof of Proposition A.2.* Fix $i \in [d]$ and let $e_i \in \mathbb{R}^d$ be the $i$-th standard basis vector. For $\mu, \sigma \in \mathbb{R}$, define a distribution $\mathcal{D}_{\mu,\sigma}$ on $\mathbb{R}$ as $\mu + \sigma \cdot e_i^T X^{-1} y$ where $X \in \mathbb{R}^{d \times d}$ and $y \in \mathbb{R}^d$ consist of independent standard Gaussians. We will denote the CDF as $\mathcal{D}_{\mu,\sigma}(< t) = \mathbb{P}\left[\mu + \sigma \cdot e_i^T X^{-1} y < t\right]$ and its complement as $\mathcal{D}_{\mu,\sigma}(> t) = \mathbb{P}\left[\mu + \sigma \cdot e_i^T X^{-1} y > t\right]$ for all $t \in \mathbb{R}$. (By Lemma A.6, $\mathcal{D}_{\mu,\sigma}$ is continuous (if $\sigma > 0$), so we do not need to worry about the strictness of the inequality.) Define $\tilde{\mathcal{D}}_{\mu,\sigma,\mathcal{R}}$ to be $\mathcal{D}_{\mu,\sigma}$ projected to $\mathcal{R}$ – i.e., to sample $\tilde{\theta} \leftarrow \tilde{\mathcal{D}}_{\mu,\sigma,\mathcal{R}}$, we sample $\theta \leftarrow \mathcal{D}_{\mu,\sigma}$ and let $\tilde{\theta} = \arg\min_{\bar{\theta} \in \mathcal{R}} |\bar{\theta} - \theta|$. We will denote the CDF similarly as before (although now the strictness of the inequality may matter).

We must reason about the impact of rounding to $\mathcal{R}$: Since $\mathcal{R} = \left\{-1 + 2\frac{i-1}{r-1} : i \in [r]\right\}$, we have

$$\text{if } t > -1, \text{ then } \mathcal{D}_{\mu,\sigma}\left(< t - \frac{1}{r-1}\right) \leq \tilde{\mathcal{D}}_{\mu,\sigma,\mathcal{R}}(< t)$$

$$\text{and}$$

$$\text{if } t < 1, \text{ then } \mathcal{D}_{\mu,\sigma}\left(> t + \frac{1}{r-1}\right) \leq \tilde{\mathcal{D}}_{\mu,\sigma,\mathcal{R}}(> t).$$

This is because the rounding can only move points by $\frac{1}{r-1}$ (unless those points are outside the interval $[-1, +1]$). Note that, if $t \leq \mu$, then $\mathcal{D}_{\mu,\sigma}(< t) \leq \frac{1}{2}$ and, similarly, if $t \geq \mu$, then $\mathcal{D}_{\mu,\sigma}(> t) \leq \frac{1}{2}$.

We run Algorithm 1 with the parameters $\varepsilon$, $d$, $\ell = 1$, and $\mathcal{R}$. We assume the input has the distribution given in the statement of Theorem 1.1. That is, independently for each $i$, we have $x_i \leftarrow \mathcal{N}(0, I)$ and $y_i = \langle \theta^*, x_i \rangle + \sigma \cdot z_i$ for $z_i \leftarrow \mathcal{N}(0, 1)$.

Let $\Theta$ be as constructed in Algorithm 1. Then the multiset $\Theta_i = \{\theta_i : \theta \in \Theta\}$ consists of $m = \lfloor n/d \rfloor$ independent samples from the distribution $\tilde{\mathcal{D}}_{\theta_i^*,\sigma,\mathcal{R}}$ defined above.

By Lemma A.3, with probability at least $1 - \beta$, we have

$$\max\left\{\left|\left\{\theta \in \Theta : \theta_i < \hat{\theta}_i\right\}\right|, \left|\left\{\theta \in \Theta : \theta_i > \hat{\theta}_i\right\}\right|\right\} \leq \left\lfloor\frac{|\Theta|}{2}\right\rfloor + \frac{2\ell d}{\varepsilon} \log\left(\frac{|\mathcal{R}|}{\beta}\right).$$

Note $|\Theta| = \ell \cdot m = m$. By Lemma A.4, with probability at least $1 - \beta$, we have

$$\left|\frac{1}{m}\left|\left\{\theta \in \Theta : \theta_i < \hat{\theta}_i\right\}\right| - \tilde{\mathcal{D}}_{\theta_i^*,\sigma,\mathcal{R}}(< \hat{\theta}_i)\right| \leq \sqrt{\frac{\log(C/\beta)}{2m}}$$

and, similarly, with probability at least $1 - \beta$

$$\left|\frac{1}{m}\left|\left\{\theta \in \Theta : \theta_i > \hat{\theta}_i\right\}\right| - \tilde{\mathcal{D}}_{\theta_i^*,\sigma,\mathcal{R}}(> \hat{\theta}_i))\right| \leq \sqrt{\frac{\log(C/\beta)}{2m}},$$

where $C$ is some universal constant. Applying a union bound, we have, for all $\beta > 0$,

$$\mathbb{P}\left[\max\left\{\tilde{\mathcal{D}}_{\theta_i^*,\sigma,\mathcal{R}}(< \hat{\theta}_i), \tilde{\mathcal{D}}_{\theta_i^*,\sigma,\mathcal{R}}(> \hat{\theta}_i)\right\} \leq \frac{1}{2} + \frac{2d}{\varepsilon m} \log\left(\frac{|\mathcal{R}|}{\beta}\right) + \sqrt{\frac{\log(C/\beta)}{2m}}\right] \geq 1 - 3\beta.$$

It follows that

$$\mathbb{P}\left[\max\left\{\mathcal{D}_{\theta_i^*,\sigma}\left(< \hat{\theta}_i - \frac{1}{r-1}\right), \mathcal{D}_{\theta_i^*,\sigma}\left(> \hat{\theta}_i + \frac{1}{r-1}\right)\right\} \leq \frac{1}{2} + \frac{2d}{\varepsilon m} \log\left(\frac{r}{\beta}\right) + \sqrt{\frac{\log(C/\beta)}{2m}}\right] \geq 1 - 3\beta.$$

The last step in the proof is to convert this bound on the quantile into an accuracy bound. Lemma A.6 tells us that the center of the distribution is at $\theta_i^*$ – i.e., $\mathcal{D}_{\theta_i^*,\sigma}(< \theta_i^*) = \mathcal{D}_{\theta_i^*,\sigma}(> \theta_i^*) = \frac{1}{2}$ – and that the distribution is roughly as concentrated around this point as a Gaussian with variance $O(\sigma)$. In particular, if $t \geq \theta_i^*$, then

$$\mathcal{D}_{\theta_i^*,\sigma}(< t) \geq \frac{1}{2} + \frac{1}{2} \mathbb{P}_{g \leftarrow \mathcal{N}(0,1)}\left[0 \leq g \leq \frac{t - \theta_i^*}{8\sqrt{C}\sigma}\right] = \frac{1}{2} + \Omega\left(\frac{t - \theta_i^*}{\sigma}\right),$$

576 where $C$ is the universal constant from Lemma A.5. Similarly, if $t \leq \theta_i^*$, then

$$\mathcal{D}_{\theta_i^*,\sigma}(> t) \geq \frac{1}{2} + \frac{1}{2}\mathbb{P}_{g\leftarrow\mathcal{N}(0,1)}[0 \leq g \leq \frac{\theta_i^* - t}{8\sqrt{C}\sigma}] = \frac{1}{2} + \Omega\left(\frac{\theta_i^* - t}{\sigma}\right).$$

577 Combining these inequalities gives

$$\max\left\{\mathcal{D}_{\theta_i^*,\sigma}\left(< \hat{\theta}_i - \frac{1}{r-1}\right), \mathcal{D}_{\theta_i^*,\sigma}\left(> \hat{\theta}_i + \frac{1}{r-1}\right)\right\} \geq \frac{1}{2} + \frac{1}{2}\mathbb{P}_{g\leftarrow\mathcal{N}(0,1)}[0 \leq g \leq \frac{|\hat{\theta}_i - \theta_i^*| - \frac{1}{r-1}}{8\sqrt{C}\sigma}]$$

$$= \frac{1}{2} + \Omega\left(\frac{|\hat{\theta}_i - \theta_i^*| - \frac{1}{r-1}}{\sigma}\right).$$

578 This rearranges to

$$|\hat{\theta}_i - \theta_i^*| \leq \frac{1}{r-1} + \sigma \cdot O\left(\max\left\{\mathcal{D}_{\theta_i^*,\sigma}\left(< \hat{\theta}_i - \frac{1}{r-1}\right), \mathcal{D}_{\theta_i^*,\sigma}\left(> \hat{\theta}_i + \frac{1}{r-1}\right)\right\} - \frac{1}{2}\right).$$

579 Combining with the high probability bound establishes

$$\mathbb{P}\left[|\hat{\theta}_i - \theta_i^*| \leq \frac{1}{r-1} + \sigma \cdot O\left(\frac{2d}{\varepsilon m}\log\left(\frac{r}{\beta}\right) + \sqrt{\frac{\log(C/\beta)}{2m}}\right)\right] \geq 1 - 3\beta.$$

580 To obtain the stated result, we simply take a union bound over all $i \in [d]$ and simplify the constants.

581 □