# OpenReview forum: "Differentially Private Linear Regression via Medians"
_NeurIPS.cc/2022/Conference — NeurIPS 2022 Submitted_

### Official Review · Reviewer_qDNE · 2022-07-10

**Rating:** 5
**Confidence:** 3
**Soundness:** 3 good
**Presentation:** 3 good
**Contribution:** 2 fair

**Summary:**

This paper studies differentially private linear regression with known covariance for isotropic Gaussian data. The algorithm is based on the median-of-mean scheme and Theil-Sen estimator. It first splits the data into different partitions, then solves the solutions for each partition, and computes the univariate median for each coordinate of the solutions using the exponential mechanism. For approximate DP, the extra cost of privacy in terms of infinity norm is $O(d^{1.5}/\varepsilon n)$, which has a $O(d^{0.5})$ gap to the lower bound. The primary advantage of this algorithm is that it does not need any clipping, which is more practical. Empirically, this paper provides extensive experiments and shows better performance for d=10, 20, 30 compared to a few baselines.

**Questions:**

1. Why is the size of each partition chosen as d?

2. Theorem 1 runs algorithm 1.1 with $\ell=1$. How does different choices of $\ell$ affect the utility guarantees?

**Limitations:**

This paper is theoretical and does not have a direct negative societal impact.

**Strengths And Weaknesses:**

Strengths: 1. This algorithm is simple and practical.
2. The presentation of this paper is clear. The structure is easy to follow.


Limitations: 1. Compared to prior works, for example [VJT22], this algorithm is not optimal. 1) Under the euclidian norm and gaussian data, the algorithm in [VJT22] has extra privacy cost $O(d/(\varepsilon n))$. The extra cost here is $O(d^{1.5}/\varepsilon n)$ in infinity norm. 2) This paper requires prior knowledge of the covariance of the Gaussian. 3) The loss is for infinity norm, which has $d^{0.5}$ gap to the euclidean norm. 4) In the experiments, this paper does not compare the algorithm with [VJT22], which shouldn't be too hard to implement.

2. Similar ideas have been used for mean estimation with sub-gaussian rates [1], DP mean estimation [2], linear regression[3]. It seems that, as a byproduct of the median of mean scheme, these types of algorithms could also provide provable robustness guarantees against corruption of the data. This is not discussed in this paper. Also, it would be better if the authors could add some related works in these topics.


[1] Hopkins, S. B. (2020). Mean estimation with sub-Gaussian rates in polynomial time. The Annals of Statistics, 48(2), 1193-1213.
[2] Hopkins, S. B., Kamath, G., & Majid, M. (2022, June). Efficient mean estimation with pure differential privacy via a sum-of-squares exponential mechanism. In Proceedings of the 54th Annual ACM SIGACT Symposium on Theory of Computing (pp. 1406-1417).
[3] Depersin, J. (2020). A spectral algorithm for robust regression with subgaussian rates. arXiv preprint arXiv:2007.06072.

---

> ### Author Response · Authors · 2022-08-02
> **Response**
>
> We thank the reviewer for their feedback and suggestions. We respond to their comments:
>
>  - *“Compared to prior works, for example [VJT22], this algorithm is not optimal.” “ In the experiments, this paper does not compare the algorithm with [VJT22], which shouldn't be too hard to implement.”*
>
> This is correct. Our result for Gaussian data is not asymptotically optimal. We did not compare to VJT22 because we were not aware of it until late in the process (it only appeared publicly in mid July). There is no implementation available. In principle, we could implement it, but there are a lot of hyperparameters/design choices in that algorithm which would complicate any experimental evaluation.
>
>
>  - *“This paper requires prior knowledge of the covariance of the Gaussian."*
>
> Our algorithm does not require knowing the data distribution. We only analyzed isotropic Gaussian data, but the algorithm would work even for non-isotropic data. Of course, the performance might degrade. Note that this issue arises even in the non-private setting, as the covariance matrix could become ill-conditioned.
>
>
>  - *“Similar ideas have been used for mean estimation with sub-gaussian rates [1], DP mean estimation [2], linear regression[3]. It seems that, as a byproduct of the median of mean scheme, these types of algorithms could also provide provable robustness guarantees against corruption of the data. This is not discussed in this paper. Also, it would be better if the authors could add some related works in these topics.”*
>
> Thank you for the suggestion, we will add some discussion of this connection and these related works. Our approach is inspired by robust statistics and we should further emphasize the connection.
>
>
>  - *“Why is the size of each partition chosen as d?”*
>
> In our experiments this gave the best performance. We tried larger size (e.g., 2d). Interestingly, we can prove better theoretical results for larger size partitions (2d).
>
> There is a tradeoff here – larger partitions mean each estimate is more accurate, but there are fewer partitions for the private median algorithm to work with. Empirically the error of the private median algorithm dominates so having more partitions is a win. Asymptotically doubling the partition size to 2d is only a constant factor loss (which is hidden by the big O notation), but we can improve the per-partition accuracy by a poly(d) factor. So the asymptotic results point to a different setting of parameters vis a vis the empirical results.
>
>
>  - *“Theorem 1 runs algorithm 1.1 with $\ell=1$. How does different choices of $\ell$ affect the utility guarantees?”*
>
> Theoretically we can analyze larger $\ell$; it makes the proof slightly more complicated. It gives the same final guarantee, so there is no win here. (The number of samples we feed to the exponential mechanism increases by a factor of $\ell$, but this is balanced out by the sensitivity increasing by the same factor. The non-private error bound also doesn’t improve because although we have more samples they are not independent.) Empirically larger $\ell$ yields slightly better results.

---

> > ### Comment · Reviewer_qDNE · 2022-08-07
> > **Thanks for your response.**
> >
> > I have read the other reviewers' comments and the authors' rebutal. Thanks to the authors for their feedback. Most of my concerns are addressed. Regarding Reviewer bc18's comments, I would like to provide more details on distributional assumptions and clarification on the theoretical guarantees.
> >
> >  > Our algorithm does not require knowing the data distribution. We only analyzed isotropic Gaussian data, but the algorithm would work even for non-isotropic data. Of course, the performance might degrade. Note that this issue arises even in the non-private setting, as the covariance matrix could become ill-conditioned.
> >
> >    1. In the non-private setting, for sub-gaussian data without knowing any information about covariance, both gradient descent and OLS solution would give you the optimal error, which is $\||\hat{\theta}-\theta^\*\||_\Sigma\simeq \sigma \sqrt{\frac{d}{n}}$. Note that this error does not scale with condition number $\kappa$ under $\Sigma$-norm. In the non-private setting, even if the covariance is ill-conditioned, the performance will not necessarily degrade.
> >
> >    2. In the private setting, we first consider the non-private error. For sub-gaussian/sub-exponential data without knowing the covariance, both [VJT22] and [LKO21] provide algorithms that achieve optimality, $\||\hat{\theta}-\theta^\*\||_\Sigma\simeq \sigma \sqrt{\frac{d}{n}}$.  For the private error, under the same settings, [VJT22] gives $\||\hat{\theta}-\theta^\*\||_\Sigma \lesssim \sigma\frac{\kappa d}{\varepsilon n}$.  [LKO21] gives algorithm that achieves optimality, $\||\hat{\theta}-\theta^\*\||_\Sigma \lesssim \sigma\frac{d}{\varepsilon n}$. This means in the private setting, even in the ill-conditioned case, knowing the covariance or condition number is not fundamentally necessary.
> >
> >    3. Reviewer bc18 has concerns about the distributional assumptions. I would like to provide more details here. For norm bounded data,  [CWZ19] and [CWZ20] (listed below) achieve the optimality $\||\hat{\theta}-\theta^\*\||_\Sigma \simeq \sigma\sqrt{\frac{d}{n}}+\sigma\frac{d}{\varepsilon n}$ for linear regression and generalized linear model.  [LKO21] achieve nearly optimal theoretical guarantees under hypercontractive distributions, sub-gaussian distributions, and even heteroscedastic settings ($x_i$ and regression noise are correlated).  As discussed above, Gaussian assumption or known covariance is not fundamentally needed to obtain such error bounds. I agree that Gaussian is natural and standard and the assumption in this paper could be potentially relaxed. But theoretically, the proposed algorithm gives $\||\hat{\theta}-\theta^\*\||_\infty \lesssim \sigma\frac{d}{\varepsilon n}$ for isotropic Gaussians, which is strictly weaker than both [LKO21] and [VJT22] under the same settings. Such theoretical comparisons with prior works are missed in the current version.
> >
> > As an empirical paper, I agree that the proposed algorithm is more practical than [LKO21]. But in my opinion, whether it is more practical than [VJT22] is still debatable without further experiments.
> >
> >
> >
> >
> > References:
> >
> > [CWZ19'] The Cost of Privacy: Optimal Rates of Convergence for Parameter Estimation with Differential Privacy by T. Tony Cai, Yichen Wang, Linjun Zhang.
> >
> > [CWZ20'] The Cost of Privacy in Generalized Linear Models: Algorithms and Minimax Lower Bounds by T. Tony Cai, Yichen Wang, Linjun Zhang.

---

> > > ### Author Response · Authors · 2022-08-09
> > > **Thank you for the helpful comments.**
> > >
> > > Thank you for your helpful comments. We will definitely add detailed discussion about this quantitative comparison to the paper. We wish to emphasize that our techniques are quite different from the prior work, which we think is of independent interest, even if the asymptotic guarantees are suboptimal.
> > >
> > > A few remarks:
> > >
> > > The non-private bound $\\|\hat{\theta}-\theta^{\*}\\|\_{\\Sigma} \\lesssim \\sigma \\sqrt{\\frac{d}{n}}$ still depends on the data distribution in two ways: First, it is in terms of the Mahalanobis norm $\\|\hat\theta-\theta^\*\\|\_\Sigma = \sqrt{(\hat\theta-\theta^\*)^T\Sigma^{-1}(\hat\theta-\theta^\*)}$, which depends on the data distribution. To convert this bound into the standard Euclidean norm we must bound the eigenspectrum of $\Sigma$.
> > > Second, the $\sigma$ on the right hand side also depends on the data distribution.
> > >
> > > Regarding the comparison to VJT22: That is a purely theoretical paper and there is no implementation available to compare against. In any case, that work appeared *after* the NeurIPS deadline. The authors shared a draft with us beforehand, but we did not have time to implement their algorithm, which would require significant optimization to make it practical.

---

> > > > ### Comment · Reviewer_qDNE · 2022-08-09
> > > > **Thanks for your response**
> > > >
> > > > I agree with the authors that the main contribution is to provide a practical DP linear regression algorithm. I understand that [VJT22] is publicly available after the NeurIPS deadline. I am just trying to provide more context and details for other reviewers from a theoretical perspective. Even if [VJT22] is more practical, this paper still provides a different way of looking at this problem. I want to emphasize that indeed the minimax error rates always depend on the data assumptions. However, knowing the covariance or not can be a critical assumption in practice from an algorithmic perspective. And this is not relevant to the error rate no matter if it would scale with the condition number.  From an algorithmic perspective, the prior works I listed do not need to know any information about the covariance, and the error bound does not scale with the condition number of covariance. However, this paper requires knowing the covariance of the distribution to get such error bounds. (Identity covariance is equivalent to knowing the covariance because you can whiten the data.) I am curious about this because there is a similar approach by [Dep20] for heavy-tailed regression with sub-gaussian rate. The connection is that both algorithms are using Median of Mean for linear regression. The difference is how the median is computed. In [Dep20], covariance is needed because the way they compute median requires prior knowledge of covariance. But here I don't think the coordinate-wise median needs to know the covariance. There are two orthogonal questions. 1. If the covariance is unknown, does the proposed algorithm have any error bounds guarantees? 2. If it does, do the bounds scale with condition number? For the first question, as an educated guess, I think the proposed algorithm will still work. Because the infinity norm measures the error in each coordinate. I think both questions can be and should be justified theoretically.  Currently, this paper claims in line 124 that covariance has to be known but without providing reasons why the algorithm would fail if the covariance is unknown.
> > > >
> > > > Remarks on the error metric and error rate: I do not think there is any standard in choosing which norm to use for linear regression. For example [RWY09] considers linear regression over $\ell_q$ ball. But typically, for linear regression, our objective is to minimize $\sum_i^n(y_i-\theta^\top x_i)^2$. It is not hard to show that this is equivalent to minimizing $\||\theta-\theta^\*\||_\Sigma$. (Minimizing excess risk is equivalent to minimizing $\Sigma$-norm). Note that this is not mahalanobis distance. Because $\||\theta-\theta^\*\||_\Sigma:= \||\Sigma^{1/2}(\theta-\theta^\*)\||$. And Mahalanobis distance is  $\||\Sigma^{-1/2}(\theta-\theta^\*)\||$. I would say $\Sigma$-norm is more "standard" for linear regression. For the euclidean norm,   I think gradient descent would also give you the optimal solution, $\||\theta-\theta^\*\||=\sigma\sqrt{d/n}$. For the private setting, I guess if you use regression depth together with exponential mechanism, you can get $\||\theta-\theta^\*\||\lesssim \frac{d\sigma}{\varepsilon n}$.  This does not require any bound on the eigenspectrum of $\Sigma$. But of course, this is inefficient.
> > > >
> > > >
> > > > [RWY09] Minimax rates of estimation for high-dimensional linear regression over $\ell\_q $-balls by Raskutti, Garvesh and Wainwright, Martin J and Yu, Bin
> > > >
> > > > [Dep20] A spectral algorithm for robust regression with subgaussian rates by Jules Depersin

---

> > > > > ### Author Response · Authors · 2022-08-09
> > > > > **Unknown covariance**
> > > > >
> > > > > > 1. If the covariance is unknown, does the proposed algorithm have any error bounds guarantees?
> > > > >
> > > > > Thanks for the great question!
> > > > >
> > > > > We can modify Lemma A.6 (in a black-box manner) to work for a non-spherical design matrix. If the data comes from $N(0,\Sigma)$ instead of $N(0,I)$, this is equivalent to replacing $X$ with $X \Sigma^{1/2}$. So the Lemma must analyze $u^T \Sigma^{-1/2} X^{-1} y$. Since $X$ is spherically symmetric, the only relevant quantity is the length of $u^T \Sigma^{-1/2}$. Looking at the proof of Proposition A.2, the final error bound will simply scale linearly with the length of this vector.
> > > > >
> > > > > One bound is $\\|u^T \Sigma^{-1/2}\\|\_2 \le \\|u\\|\_2 \cdot \\|\Sigma^{-1/2}\\|\_{\text{op}} = 1 / \sqrt{\lambda\_{\text{min}}(\Sigma)}$. So the error of our algorithm can be bounded in terms of the smallest eigenvalue of the covariance.
> > > > >
> > > > > However, we can do a bit better:  For the $i$-th coordinate, the error scales with $\\|u^T \Sigma^{-1/2}\\|_2$ where $u=e_i$ is the $i$-th standard basis vector. If we look at the squared error summed over all the coordinates, this scales with $$\sum_i^d \\|e\_i^T \Sigma^{-1/2}\\|\_2^2 = \sum_i^d e_i^T \Sigma^{-1} e_i = \mathsf{trace}(\Sigma^{-1}) = \\|\Sigma^{-1/2}\\|\_{\text{F}}^2 = \sum\_i^d 1/\lambda_i(\Sigma).$$
> > > > >
> > > > > **In summary, for an unknown covariance $\Sigma$ of the Gaussian features, we can bound the infinity norm of the error in terms of $1 / \sqrt{\lambda\_{\text{min}}(\Sigma)}$ and the 2-norm of the error in terms of $\sqrt{\sum\_i^d 1/\lambda\_i(\Sigma)}$.**
> > > > >
> > > > > > 2. If it does, do the bounds scale with condition number?
> > > > >
> > > > > These quantities $1 / \sqrt{\lambda\_{\text{min}}(\Sigma)}$ and $\sqrt{\sum\_i^d 1/\lambda\_i(\Sigma)}$ are not quite the condition number, but they are closely related.
> > > > >
> > > > > You will note that if we double the covariance $\Sigma \mapsto 2\Sigma$ this has no effect on the condition number, but it will reduce the error bounds above by a factor of $1/\sqrt{2}$. This is the correct behaviour because, if we scale up the features while holding the noise scale $\sigma$ fixed, then, intuitively, the ratio of signal to noise is increased and we can obtain more accurate estimates of the parameters.
> > > > >
> > > > > Of course, if we use excess risk or, equivalently, $\\|\hat\theta-\theta^\*\\|\_\Sigma$ (thanks for the correction -- this is not quite the Mahalanobis norm) as our metric then we avoid this scaling issue. We decided to give a guarantee in terms of $\\|\hat\theta-\theta^\*\\|\_\infty$ because it makes sense for our algorithm and, from a statistics perspective, estimating the parameters is the end goal.
> > > > >
> > > > > It would be a very interesting question to obtain a practical linear regression algorithm whose error guarantees adapt to the covariance. To the best of our knowledge, even for the simpler task of mean estimation this is still unresolved. (Although, like for linear regression, we have theoretical results showing this is possible.)

---

### Official Review · Reviewer_mEFt · 2022-07-10

**Rating:** 5
**Confidence:** 3
**Soundness:** 3 good
**Presentation:** 4 excellent
**Contribution:** 2 fair

**Summary:**

This paper addresses the problem of differentially private linear regression. Specifically, current DP methods need access to the sensitivity of the query which is in practice hard to compute. Alternative methods clip the data, harming performance. This paper proposes a method that relies on privately computing a multidimensional “median”, a good estimator for the mean under certain assumptions and that has smaller sensitivity.

**Questions:**

- How to compute range R in real applications? if R is set too large, then the exponential mechanism will add too much mass to small depth points, and if it is too small, the range may not even contain the real model.
- How does the restriction in line 80 compares to clipping?
- Why use the max instead of the min in the score function? Wouldn't this assign very high utility to a point in the boundary?
- Why is $\sigma$ described as a parameter? I don’t see it used for any pre-processing or actual algorithm step.
- Can you discuss the choice of hyperparameters for DP-GD?
- Can you discuss the results in figure 5? why does the Widened Exponential from [AMSSV22] degrade in lower dimensions?


**Limitations:**

- The paper motivates well the use of a "high dimensional private median" for DP linear regression. However, it is not clear how this will overcome the challenges that current methods face, like clipping, since setting a range R is basically finding a clipping rate.
- Step 15 for calculating the median could result in a point outside the closure of datapoints. A common example for this is having points (1,0,0), (0,1,0), and (0,0,1). Taking the median along each axis results in (0,0,0) which is outside the plane defined by the three points.
- Experimental section is based on synthetic data that follows the assumptions on theorem and thus provides guidance on how to select the parameters, however, realistic datasets may behave very differently.
- Experimental section only compares to one set of fixed parameters of DP-SGD. Perhaps following guidance from theory or discussing the choice of hyperparameters for DP-GD could justify this choice.

In general, the paper proposes an alternative approach but that still requires an equivalent to clipping, and this is not addressed neither theoretically nor empirically.

**Strengths And Weaknesses:**

*Strengths*
- Nice analysis and intuition for Gaussian data.
- Very clear and organized.
- Good related work and comparison section.


*Weaknesses*:
- Missing relevant recent work on DP-medians/quantiles.
- Experiments only test on synthetic data that meets the assumption. However, finding a reasonable r (even in small settings) could be hard.
- Data complexity could be rather large, and prohibitive in real settings where linear regression datasets are rather small.
- The high dimensional median proposed in step 15 may not even be an interior point in the convex closure of the data points.

---

> ### Author Response · Authors · 2022-08-02
> **Response**
>
> We thank the reviewer for their time and comments. We respond to the main points:
>
>  - *“Missing relevant recent work on DP-medians/quantiles.”*
>
> Are you referring to Kaplan, Schnapp, & Stemmer (ICML 2022)? We will add this reference. In our application we are only looking for the median, rather than multiple quantiles, so it doesn’t seem like this paper would improve over the exponential mechanism. But multiple quantiles would be relevant if, e.g., we want to output confidence intervals for the coefficients.
>
>  - *“Experiments only test on synthetic data that meets the assumption. However, finding a reasonable r (even in small settings) could be hard.” “How to compute range R in real applications? if R is set too large, then the exponential mechanism will add too much mass to small depth points, and if it is too small, the range may not even contain the real model.”*
>
> The advantage of our method is that it is very insensitive to the parameter $r$/$\mathcal{R}$. Asymptotically the dependence is only logarithmic. In practice, extending $\mathcal{R}$ has negligible impact because the probability of outputting a point far outside the true range is exponentially small.
>
>  - *“How does the restriction in line 80 compares to clipping?”*
>
> The paper cited on line 80 shows that the number of samples must grow with the iterated logarithm of our parameter $r$. This is an *extremely* slow-growing function and practically constant. But it shows that we cannot ignore this parameter entirely.
>
> We use the exponential mechanism because it is practical. But it is asymptotically suboptimal. There are algorithms whose asymptotic sample complexity is polynomial in the iterated logarithm of $|\mathcal{R}|$, but these algorithms are far from practical.
>
> In contrast, clipping introduces a harsh privacy-utility tradeoff. If the clipping bounds are loose by a factor of 2, then we introduce twice as much noise as necessary. That is, the error grows linearly with the clipping, but our algorithm only has a logarithmic dependence on the range. On the other hand, if the clipping bound is too tight, this distorts the data; the effects of this on the final output are not fully understood.
>
>  - *“Why is $\sigma$ described as a parameter? I don’t see it used for any pre-processing or actual algorithm step.”*
>
> Indeed, $\sigma$ is not a parameter of the algorithm. If we described it as a parameter, that should be corrected.
>
>  - *“Can you discuss the choice of hyperparameters for DP-GD?”*
>
> Our version of DP-GD regressor has three hyperparameters: clipping norm, number of epochs, and learning rate. For the clipping norm we used the lipshitz constant of the gradient (it is known since the feature distribution is bounded) and made some attempt to manually find the best number of epochs and learning rate. In the plots we use learning rate equal to 0.1 and the number of epochs equal to 100.
>
>  - *“Can you discuss the results in figure 5? why does the Widened Exponential from [AMSSV22] degrade in lower dimensions?”*
>
> According to our experiment, there is no degradation of the Widened Exponential in lower dimensions. Rather when the number of samples is not sufficient both algorithms are performing bad and for higher dimensions we need more samples, so the cases where the regular exponential mechanism start performing better is not shown on the figure.
>
>  - *“The paper motivates well the use of a "high dimensional private median" for DP linear regression. However, it is not clear how this will overcome the challenges that current methods face, like clipping, since setting a range R is basically finding a clipping rate.”*
>
> The reviewer is correct that our algorithm does not entirely escape the need to bound the range of the data; indeed, it is impossible to completely avoid bounding the range of the data. However, our methods improve the dependence on this parameter from linear to logarithmic, which we consider to be a significant improvement both in theory and practice.
>
>  - *“Step 15 for calculating the median could result in a point outside the closure of datapoints.”*
>
> Indeed, taking a coordinate-wise median is not ideal. Ideally, we would compute something like a Tukey median, but this is difficult to compute non-privately, yet alone with DP. Our design choice here is motivated by practicality. The exponential mechanism is also not asymptotically optimal, but it seems to be the most practical method for computing medians.
>
>  - *“Experimental section is based on synthetic data that follows the assumptions on theorem and thus provides guidance on how to select the parameters, however, realistic datasets may behave very differently.”*
>
> We are working to add results with real datasets. Realistic datasets tend to be heavier-tailed than synthetic datasets, which is actually an advantage of our approach compared to clipping.

---

> > ### Comment · Reviewer_mEFt · 2022-08-09
> > **Thanks for the response.**
> >
> > I thank the authors for the time spent on clearly answering all questions.
> >
> > I understand better now the improvement on this paper vs. clipping and it does seems like a bad range R is less harmful than a bad clipping rate.
> >
> > It seems that several claims could be confirmed with either theoretical results with assumptions that are less restrictive, like data being Gaussian. Further,  given the fact that there are some heuristics involved in the algorithm, claims on the success of heuristics should be supported on real or broader classes of distributions.

---

### Official Review · Reviewer_bc18 · 2022-07-11

**Rating:** 4
**Confidence:** 4
**Soundness:** 3 good
**Presentation:** 2 fair
**Contribution:** 2 fair

**Summary:**

The paper studies the problem of differentially private linear regression and develops a new algorithm for this problem based on privately calculating 1-dimensional medians. The algorithm is based on the Theil-Sen estimator and uses the exponential mechanism to privately estimate the medians required for this estimator. The authors present a utility guarantee for their algorithm with Gaussian features and Gaussian noise. Moreover, they provide some experiments over synthetic datasets that compare their algorithm to existing algorithms.

**Questions:**

No

**Limitations:**

Yes

**Strengths And Weaknesses:**

The paper studies an important problem (private linear regression). However, I think the authors need to present more theoretical results, comparison to prior work, and experimental evidence to make this paper more compelling.

Weaknesses
1.	The main (and only) theoretical result in the paper provides utility guarantees for the proposed algorithm only when the features and noise are Gaussian. This is a strong requirement on the data, especially given that previous algorithms don’t need this assumption as well. Moreover, the authors should compare the rates achieved by their procedure to existing rates in the literature.
2.	Experiments: the experimental results in the paper don’t provide a convincing argument for their algorithms. First, all of the experiments are done over synthetic data. Moreover, the authors only consider low-dimensional datasets where d<30 and therefore it is not clear if the same improvements hold for high-dimensional problems. Finally, it is not clear whether the authors used any hyper-parameter tuning for DP-GD (or DP-SGD); this could result in significantly better results for DP-GD.
3.	Writing: I encourage the authors to improve the writing in this paper. For example, the introduction could use more work on setting up the problem, stating the main results and comparing to previous work, before moving on to present the algorithm (which is done too soon in the current version).


More:

1.	Typo (first sentence): “is a standard”
2.	First paragraph in page 4 has m. What is m? Should that be n?

---

> ### Author Response · Authors · 2022-08-02
> **Response**
>
> We thank the reviewer for their time and feedback. We respond to the main points:
>  - *“The main (and only) theoretical result in the paper provides utility guarantees for the proposed algorithm only when the features and noise are Gaussian. This is a strong requirement on the data, especially given that previous algorithms don’t need this assumption as well.”*
>
> Gaussian data is a very natural and standard assumption. The assumption could be relaxed – e.g., we could instead assume bounds on the norms of the design matrices and error vectors – but this would make the result harder to interpret without really providing additional insight.
>
> What kind of result would the reviewer like to see? If the reviewer has a particular data model in mind, we can try to analyze it.
>
> We emphasize that some kind of assumption on the data is necessary to give these kinds of bounds, even non-privately. So it is not true that previous algorithms don’t have any assumptions, although it could take a different form such as bounding the condition number of the covariance matrix; but such a bound is best justified/explained via distributional assumptions.
>
> - *“Experiments: the experimental results in the paper don’t provide a convincing argument for their algorithms. First, all of the experiments are done over synthetic data.“*
>
> We are working to add results with standard “real” datasets. If the reviewer has any suggestions for particular datasets that we should consider, those would be appreciated.
>
> We used synthetic data because it provides an apples-to-apples comparison between methods, as we can ensure that the data is indeed bounded (i.e. no clipping required).  In particular, the methods we compare to require a priori bounds on the data, and computing such bounds is a challenge in practice.
>
>
>  - *“Moreover, the authors only consider low-dimensional datasets where d<30 and therefore it is not clear if the same improvements hold for high-dimensional problems.”*
>
> We will extend the plots to higher dimensions. Is there a particular value of d that would be of interest?
>
>
>  - *“Finally, it is not clear whether the authors used any hyper-parameter tuning for DP-GD (or DP-SGD); this could result in significantly better results for DP-GD. “*
>
> We made some attempt to manually optimize the DP-GD parameters, but the reviewer is correct that there may be further room for improvement via an exhaustive hyperparameter search. We note that setting hyperparameters is a significant challenge in practice, which is a limitation of DP-GD.
>
> We also remark that we used DP-GD instead of DP-SGD so that we can obtain tight privacy bounds via Gaussian DP, whereas DP-SGD requires subsampling which yields a suboptimal privacy-utility tradeoff. The downside of DP-GD is that it is quite slow, which was a major bottleneck in our experiments.
>
>
>  - *“I encourage the authors to improve the writing in this paper. For example, the introduction could use more work on setting up the problem, stating the main results and comparing to previous work, before moving on to present the algorithm (which is done too soon in the current version).”*
>
> We will work to improve our manuscript, in particular by adding further discussion of prior work. We are surprised by the reviewer’s comment that the algorithm is presented too soon; linear regression is such a well-known problem that we feel it needs little introduction. If there is anything that is unclear, we would appreciate this being pointed out.
>
>
>  - *“First paragraph in page 4 has m. What is m? Should that be n?”*
>
> We are not sure which paragraph the reviewer is referring to, but $m = \lfloor n / d \rfloor$ is the number of partitions (defined in Algorithm 1) and $n$ is the total number of samples.

---

### Meta-Review · Area_Chair_QUL6 · 2022-08-26

**Recommendation:** Reject
**Confidence:** Certain

**Metareview:**

Though the reviewers appreciate the contribution overall, and the application of median methods to regression for the purpose of avoiding/circumventing clipping is novel, the strength of the contributions remains limited in light of other existing work that achieves more favorable bounds and/or uses fewer assumptions. This is certainly the case for  (unpublished) [VJT22], but it is also the case for [MKFI22], which also assumes Gaussian inputs but, crucially, does not rely on prior knowledge of the distribution's covariance. Moreover, in light of the effort required to perform DP covariance estimation via [KLSU19], it is not clear that, as stated in the rebuttal, an error proportional to condition number is non-trivial. For example, it is not clear that, allowing for such error, processes like [KLSU19], do not become trivial.

A reorganization of the paper, that brings related work earlier, and explains what is known regarding regression in the unbounded regime (especially in light of [VJT22],[MKFI22]) but also on DP for other settings in the unbounded regime, and where the current work is placed/what the contributions beyond these works are, would strengthen the paper.



**Award:**

No

---

### Decision · Program_Chairs · 2022-09-14

Reject